# Scaling Physical Reasoning with the PHYSICS Dataset

**Shenghe Zheng**[1,2*], **Qianjia Cheng**[1,5*], **Junchi Yao**[1,6*], **Mengsong Wu**[1,7],
**Haonan He**[1,8], **Ning Ding**[1,3], **Yu Cheng**[1,4], **Shuyue Hu**[1], **Lei Bai**[1],
**Dongzhan Zhou**[1], **Ganqu Cui**[1†], **Peng Ye**[1,4†]

[1] Shanghai Artificial Intelligence Laboratory    [2] Harbin Institute of Technology
[3] Tsinghua University    [4] The Chinese University of Hong Kong    [5] Beihang University
[6] University of Electronic Science and Technology of China    [7] Soochow University
[8] University of Science and Technology of China
`shenghez.zheng@gmail.com`

## Abstract

Large Language Models (LLMs) have achieved remarkable progress on advanced reasoning tasks such as mathematics and coding competitions. Meanwhile, *physics*, despite being both reasoning-intensive and essential to real-world understanding, received limited academic and industrial attention. This paper introduces PHYSICS, a dataset containing 16,568 high-quality physics problems spanning subjects and difficulty levels, to facilitate this issue. Specifically, PHYSICS is curated with exercises from over 100 textbooks through a carefully designed pipeline for quality control. It covers five major physics domains: Mechanics, Electromagnetism, Thermodynamics, Optics, and Modern Physics. It also spans a wide range of difficulty levels, from high school to graduate-level physics courses. To utilize the data for improving and evaluating the model's physical reasoning capabilities, we split the dataset into training and test sets, and provide reasoning paths generated by powerful reasoning models for the training data to facilitate model training. In addition, for the evaluation part, we find that existing evaluation frameworks exhibit biases in aspects such as units, simplification, and precision in physics domain. To balance efficiency and accuracy, we introduce a Rule+Model evaluation framework tailored to physics problems. Our evaluations on current state-of-the-art open-source and proprietary models highlight the limitations of current models in handling physics-related tasks. We hope that our dataset and evaluation methodology will jointly advance the development of LLMs in the field of physics. The code and data can be found at: https://github.com/Zhengsh123/PHYSICS.

## 1 Introduction

The rapid expansion of reasoning capabilities and world knowledge in large language models (LLMs) has led to a sharp increase in their intelligence [9, 63, 53]. In fields such as mathematics and coding, LLMs can now handle problems at the Olympiad level, reaching or even surpassing human expert performance in some cases [15, 54, 19]. However, physics, despite being the foundation of all the natural sciences [18, 43], has not received comparable attention in the development of language models. As a result, the understanding of physics in current models remains significantly limited [5]. Given the physical nature of the real world, the ability to understand and apply physics is critical for AI to accurately model and interact with reality [10, 57]. The physical reasoning capability of LLMs ultimately determines their effectiveness in assisting humans in real-world scenarios [52, 23].

---

[*]Equal Contribution. Work done during the internship at Shanghai AI Lab.
[†]Corresponding Author.

39th Conference on Neural Information Processing Systems (NeurIPS 2025) Track on Datasets and Benchmarks.

Table 1: Comparison of physics datasets. *Level* indicates question difficulty: 1: High school and below; 2: High School Olympiad; 3: Undergraduate (Non-Physics Major); 4: Undergraduate/Postgraduate(Physics Major). *Scale* refers to dataset size. *Dataset Split* indicates whether the dataset is divided into training and test sets. *Subjects* indicates the range of disciplines covered in the dataset. For *Language*, EN stands for English, and ZH stands for Chinese. In *Eval*, Spec. indicates the use of a physics-specific evaluation method. *Leak. Det.* represents information leakage detection.

| | Level | Scale | Training/Test | Subjects | Language | Eval | Leak. Det |
|---|---|---|---|---|---|---|---|
| MMLU [21] | 1,3 | 548 | ✗ | 3 | EN | Rule | ✗ |
| AGIEval [67] | 2 | 200 | ✗ | - | ZH | Rule | ✗ |
| C-Eval [24] | 1,3 | 601 | ✗ | - | ZH | Rule | ✗ |
| GAOKAO [65] | 1 | 111 | ✗ | - | ZH | Rule+Model | ✗ |
| JEEBench [3] | 1 | 123 | ✗ | - | EN | Rule | ✗ |
| CMMLU [31] | 1,3 | 423 | ✗ | 3 | ZH | Rule | ✗ |
| SciEval [48] | - | 1657 | ✗ | 3 | EN | Rule | ✗ |
| PhysQA [14] | 1 | 1770 | ✗ | 3 | EN | Rule | ✗ |
| GPQA [47] | 3,4 | 227 | ✗ | 5 | EN | Rule | ✗ |
| OlympiadBench [19] | 2 | 376 | ✗ | 5 | EN&ZH | Rule | ✓ |
| OlympicArena [25] | 2 | 796 | ✗ | 5 | EN&ZH | Rule+Model | ✓ |
| PhysicsQA [27] | 1 | 370 | ✗ | 5 | - | Rule | ✗ |
| UGPhysics [61] | 3,4 | 11040 | ✗ | 3 | EN&ZH | Rule+Model | ✓ |
| PHYBench [44] | 1,2,3 | 500 | ✗ | 5 | EN | Spec Rule | ✓ |
| PHYSICS (ours) | **1,2,3,4** | **16568** | ✓ | 5 | **EN&ZH** | **Spec. Rule+ Spec. Model** | ✓ |

Due to limited attention from both academia and industry, large language models currently face significant challenges in developing physical reasoning capabilities, particularly in terms of **data** and **evaluation** frameworks. Regarding data [61, 44], there are two main issues: (a). **Lack of high-quality training data**. This makes it difficult to effectively enhance models' abilities in the physics vertical and limits the development of their physics understanding and reasoning skills. (b). **Imbalanced test data distribution**. Existing physics test sets often cover only a narrow range of difficulty levels and subject areas, resulting in low discriminability and limited diversity in evaluation. For evaluation, there is a **lack of dedicated evaluation frameworks**. Most current frameworks borrow metrics from mathematics [61, 44]. However, physics introduces unique evaluation challenges such as unit conversion and numerical simplification, which existing frameworks struggle to handle. We further construct a test set to quantify the evaluation errors that arise from the current framework. These limitations not only hinder a precise assessment of model performance but also limit our ability to provide accurate guidance for improving models' physical reasoning capabilities.

To address the data challenge, we introduce PHYSICS, a large-scale physics dataset with the broadest difficulty coverage to date, including both training and test splits. We curated 8,284 high-quality physics problems and solutions from over 100 carefully selected textbooks using a rigorous extraction and cleaning framework. The dataset spans five major physics domains: Mechanics, Electromagnetism, Thermodynamics, Optics, and Modern Physics, and covers difficulty levels ranging from high school to graduate-level. To enable bilingual evaluation, we further translate all problems between English and Chinese, leading to a total of 16,568 questions. For the test set, we carefully balance both difficulty and subject distributions, allowing for a comprehensive evaluation of physics capabilities across a wide range of topics and skills. For training, we set aside 14,568 samples as the training set and provide reasoning paths from powerful reasoning models to facilitate models' physics capability.

Regarding the evaluation framework optimization, to balance accuracy and efficiency, we adopt a hybrid approach that combines rule-based and model-based methods. We are the first to design a dedicated hybrid evaluation framework specifically for physics-related tasks. For issues such as unit conversion and numerical simplification mentioned above, we use predefined rules to specify transformation relationships, and fine-tune existing judge models on the training set with manual annotation. This dual improvement in rules and models enhances the evaluation framework's effectiveness. At the same time, we construct an artificially annotated test set to validate the effectiveness of this improvement. The improved framework ensures accurate and robust assessment of models' physical reasoning capabilities, which helps guide their further development.

In addition, we conduct extensive experiments on both open-source and closed-source models. We find that, in general, current open-source models still lag behind closed-source models, and reasoning

models outperform non-reasoning models. While LLMs show strong mathematical reasoning abilities, even the strongest models, such as OpenAI-o3 [41] and Gemini-2.5-pro [16], perform poorly on physics problems. These results highlight the limitations of current models in physics reasoning, pose challenges for further development, and suggest that improving physics capabilities is an important future direction for LLM advancement. In summary, our contributions are as follows:

- We construct PHYSICS, a high-quality physics dataset with the largest scale and broadest difficulty coverage to date, featuring separate training and test splits. It supports both effective training for physics capability improvement and targeted evaluation of models' physics performance.

- We propose a Rule+Model framework, the first to jointly design rules and models tailored for physics problems, enabling more accurate evaluation of physics reasoning.

- We conduct extensive physics capability evaluations across various LLMs. Our results reveal significant limitations in current models' physics performance. We provide in-depth analysis highlighting the challenges that need to be addressed for improving LLMs' physics capabilities.

## 2 Related Work

**Physics Datasets.** Efforts have been made to enhance and evaluate the intelligence of large language models by developing diverse datasets across multiple domains. Mathematics [22, 11, 15, 32, 49, 62] and coding [35, 8, 4, 26, 1] have become key domains for evaluating model performance. However, despite the importance of physics for the real world, high-quality physics datasets for training and evaluation remain limited. Existing physics data sources mainly come from two types: multi-discipline datasets [25, 24, 30, 34, 47, 56, 48] and physics-specific datasets [61, 44]. Multi-discipline datasets often contain only a small amount of physics-related content, making it difficult to fully train or evaluate models' physics capabilities. Physics-specific datasets, on the other hand, often suffer from poor quality control, limited quantity, narrow difficulty ranges, and lack of clear training-test splits. To advance the physics reasoning capabilities of LLMs, we introduce a comprehensive, text-based physics dataset designed to serve as a robust dataset for both training and evaluation.

**Evaluation Method.** Reasoning-intensive tasks [44, 19] highlight the need for advanced evaluation methods for LLMs. Rule-based approaches [20, 55, 29]ensure accuracy via domain rules but may fail in handling long reasoning contexts. Model-based evaluators [7, 68, 6] offer an alternative, but most are often designed for mathematical problems and face challenges such as **Numerical Simplification** and **Unit Conversion** when applied to physics, leading to biased results. To address these issues, we propose a Rule+Model framework tailored specifically for physics problems.

## 3 The PHYSICS Dataset

### 3.1 Overview

Our work consists of two main components: the dataset and the evaluation framework. For the dataset, we introduce PHYSICS, a large-scale, high-quality physics dataset with comprehensive difficulty coverage. To enhance models' physics reasoning abilities, PHYSICS spans five major domains, including Mechanics, Electromagnetism, Thermodynamics, Optics, and Modern Physics, and covers difficulty levels from high school to graduate-level. The dataset is built from carefully selected textbooks through rigorous filtering, quality control, and leakage detection. We provide detailed reasoning paths for the training set from powerful reasoning models and ensure balanced difficulty and subject distributions in the test set. More details can be found in Sec. 3.2, Fig. 1, and Appendix B.

Table 2: Dataset Statistic.

| Statistic | Number |
| --- | --- |
| Total Problems | 8284 |
| + Translation | 16568 |
| Total Subjects | 5 |
| Total Answer Types | 7 |
| Total Difficulty Level | 4 |
| Language (EN : ZH) | 1:1 |
| Data Split (Train : Test) | 7:1 |
| Average Problem Tokens | 122.29 |
| Average Solution Tokens | 385.16 |

As for the evaluation part, we design a specialized framework tailored to physics tasks. To balance efficiency and accuracy, we adopt a Rule+Model based evaluation approach, specifically addressing common evaluation challenges in physics answers such as unit conversion and numerical simplification. Further details are provided in Sec. 3.3 and Appendix D.

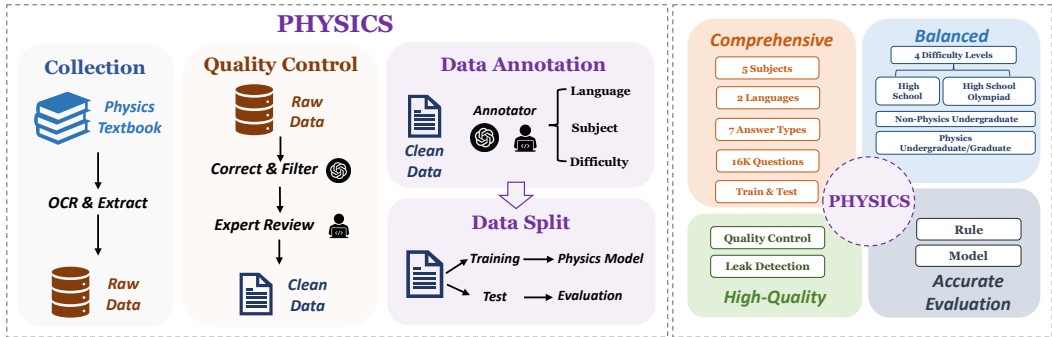

Figure 1: Pipeline of PHYSICS construction process (left) and characteristics of PHYSICS (right).

## 3.2 Dataset Construction

### 3.2.1 Data Collection

Due to the limited availability of existing physics-related datasets, we construct a comprehensive, domain-specialized, and high-quality physics dataset by extracting problems from physics textbooks and exercise books. The construction process consists of three main steps: **PDF to Markdown.** Original textbooks in PDF format are parsed into Markdown (MD) files using Optical Character Recognition (OCR) tools, enabling efficient text-based processing. **Question Extraction.** Since questions and answers are often placed in close proximity within most books, we apply a sliding window approach to traverse the Markdown documents. GPT-4o is employed to extract questions, answers, and question-answer pairs. To reduce hallucination and ensure alignment with the source material, we cross-reference each extracted pair with its original position in the document. **Matching Questions and Answers.** For materials where questions and answers are presented separately, we utilize metadata such as chapter numbers, problem indices, and formatting patterns to accurately match questions with their corresponding answers. Details can be found in Appendix B.

### 3.2.2 Quality Control

To enhance the overall quality and reliability of the extracted data, we conduct multiple cleaning procedures focused on the following aspects: **OCR Error Correction**: OCR outputs occasionally contain recognition errors due to low scan quality or ambiguous formatting, such as misidentifying the digit *3* as *5* or misinterpreting mathematical expressions. We leverage GPT-4o with contextual understanding to detect and correct such errors. **Data Filtering**: We remove multi-modal data that required visual information (e.g., images), questions that relied heavily on external text context, and question-answer pairs that could not be accurately matched. **Expert Review**: The previous two steps rely primarily on LLMs, which may introduce some errors. Therefore, in this step, human experts are introduced to further review and refine the results, removing data with issues such as incorrect question-answer extraction, mismatched pairs, or wrong answers, thereby ensuring high data quality. After data post-processing, we ultimately obtain 8,284 complete high-quality physics questions and answers. We perform Chinese-English translation on the existing data, effectively doubling the amount of data obtained. The detailed translation procedure can be found in Appendix B. The statistics of PHYSICS can be found in Tab. 2.

### 3.2.3 Data Annotation

To enable more effective use of the data, we classify the data along multiple dimensions. The main classification criteria fall into three categories: **Language:** The collected data includes both Chinese and English texts, categorized based on their source. **Difficulty Level:** We categorize questions into four difficulty levels: High School, High School Olympiad, Undergraduate (Non-Physics Major), and Undergraduate/Postgraduate (Physics Major). The statistics can be found in Fig. 2. This classification is primarily based on data sources, with some input from expert reviewers for certain items. **Subject Classification:** Each question is labeled with one of five physics subfields that include Modern Physics, Mechanics, Electromagnetism, Thermodynamics, and Optics. The statistics can be found in Fig. 3. This labeling is mainly based on the source information. For cases where subject information is unclear, we use a combination of LLM assistance and expert annotation.

Table 3: Answer types.

| Type | Example |
|------|---------|
| Interval | $[-1,1]$ |
| Expression | $4R/3\pi$ |
| Equation | $\boldsymbol{F} = \nabla(p \cdot E)$ |
| True / False | True |
| Multiple Choice | A |
| Numerical Value | $1.8 \times 10^{-4}$ |
| Open-End | $x$ remains constant. |

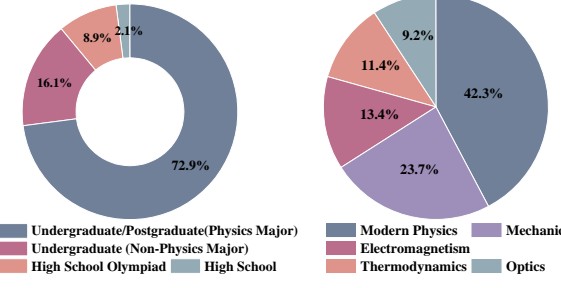

Figure 2: Difficulty Distribution of PHYSICS.

Figure 3: Subject Distribution of PHYSICS.

### 3.2.4 Data Split

To effectively improve and evaluate models' physics capabilities, we split the dataset into training and test sets. We sample 2000 items as the test set with balanced subject and difficulty coverage. The remaining data are used for training. Translation pairs from the same source are kept within the same set to avoid information leakage. For the training set, we provide detailed reasoning paths of powerful reasoning models to assist in improving models' physics reasoning abilities. The detailed construction scheme of the training set can be found in Appendix B.

### 3.2.5 Data Leakage

To prevent data leakage from affecting the evaluation on PHYSICS, we have removed any overlapping content between our collected data and existing open-source datasets during the data curation process. To further check for potential overlap between the PHYSICS test set and the training data of LLMs, we use n-gram matching for leakage detection [60]. Specifically, we randomly select positions of from each test sample. If the 5-gram predicted by the model matches the actual 5-gram, the sample is considered contaminated. In Tab. 4, we show the results for several LLMs. The results indicate that data leakage is rare with minimal impact on the evaluation.

Table 4: Data Leakage Detection. $prop_1$: leaked data proportion; $prop_2$: model accuracy on leaked data.

| Model | $prop_1$ | $prop_2$ |
|-------|----------|----------|
| Qwen3-8B [46] | 0% | 0% |
| LLaMA3.1-8B-Instruct [17] | 0% | 0% |
| Gemma2-9B [50] | 0.3% | 0% |
| DeepSeek-MOE-16B-Chat [12] | 0.5% | 40% |
| Mistral-Nemo-Instruct-2407 [38] | 0% | 0% |
| QwQ-32B [51] | 0.5% | 60% |

## 3.3 Evaluation

In this section, we introduce our evaluation framework. We find that current judgment frameworks exhibit certain biases when evaluating physics problems, as shown in Tab. 5. The main issues include: (1) **Unit conversion**. Some physical units involve dimensions, which existing methods fail to handle properly; (2) **Simplification and**

Table 5: Evaluation error cases

| Case | Ground Truth | Model Output |
|------|--------------|--------------|
| Unit Conversion | $0.6 \times 10^{-6} \, m$ | $600 \, nm$ |
| Simplification | $\frac{10^6(-1000\alpha)}{2\ln 3\omega}$ | $\frac{-4.557\cdot10^8\alpha}{\omega}$ |

**Approximation**. Various forms of simplification and approximation are common in physics, but current methods lack sufficient capability in recognizing them. To ensure accurate evaluation of model responses, we design a physics-specific evaluation approach tailored to these challenges.

**Rule+Model Framework.** A rule-only judgment method struggles to handle complex composite formulas in physics, while relying solely on model-based judgment introduces high computational cost. To balance accuracy and efficiency, we propose a Rule+Model framework, as detailed in Alg. 1. We first use rule-based judgment to assess the answer; if the rule judges it as incorrect, we then apply the model for a second check. Only when both methods judge the answer as wrong is it considered incorrect.

---

**Algorithm 1** Workflow of Rule+Model Evaluation

**Input:** Question, Ground Truth, Model Output
**Output:** Correct or Incorrect
1: Result ← Rule-Verify(input)
2: **if** $Result = Correct$ **then**
3:     **return** Result
4: **else**
5:     **return** Model-Verifier(input)
6: **end if**

---

**Physics-specific Optimization.** Since current evaluation methods fail to address the aforementioned issues such as unit conversion and simplification, we introduce targeted optimizations. For the rule-based component, we adopt the math-verify [29] as the base rule-verifier and pre-define a set of unit conversion rules, enabling automatic conversion. However, due to the limited coverage of rules and the inherent limitations of rule-based approaches, we further fine-tune an existing model-verifier. We select xVerify-8B-I [7], a model designed for mathematical reasoning judgment, as our base verifier. We first construct training and test data by using GPT-4o to generate multiple equivalent forms of physics answers, followed by human verification to ensure accuracy. To enhance diversity, we also include some mathematically equivalent responses. We then split the data into training and test sets; detailed construction procedures and statistics are provided in the Appendix D.

Based on the base verifier, we fine-tune and get a physics-specific verifier, named *physics-xVerify*. In Tab. 6, we report the accuracy and time cost of different evaluation methods on our 2k-sample test set. The results show that the combination of our *rule-verify* and *physics-xVerify* achieves a 12.58% improvement over existing methods with acceptable time cost. Details can be found in Appendix D.4. This demonstrates the effectiveness of our proposed evaluation method. This accurate evaluation framework is essential for effectively guiding the development of physical reasoning.

Table 6: Comparison of Evaluation Methods on our human-annotated test dataset.

| Model | Acc.(%) | Time |
|---|---|---|
| **Rule** | | |
| Omni-Math-Rule [15] | 34.20 | 1 min 29 s |
| Rule-Verifier | 38.62 | 1 min 46 s |
| **Rule+Model** | | |
| Rule-Verifier + GPT-4o [39] | 58.58 | 20 min 25 s |
| Rule-Verifier + Omni-Judge [15] | 67.39 | 7 min 14 s |
| Rule-Verifier + xVerify [7] | 83.34 | 4 min 28 s |
| Rule-Verifier + physics-xVerify | **95.92** | 4 min 37 s |

## 4 Experiment

In this section, we present experiments on both test and training data from the PHYSICS dataset. Results for the test set are analyzed in Sec. 4.1, and those for the training set in Sec. 4.2.

### 4.1 Evaluation Result and Analysis

For evaluation, we test both closed-source models and open-source models. These are further categorized into reasoning models, such as OpenAI-o3, and chat models, including GPT-4.1 and so on. Performance is assessed on both Chinese and English questions across five domains of our benchmark. This comprehensive evaluation framework allows us to compare the reasoning abilities of different types and scales of LLMs in multilingual and multi-domain settings. Detailed evaluation settings can be found in Appendix C. Tab. 7 summarizes the evaluation results of various models on our test set. Next, we analyze the experimental results.

**Despite their impressive capabilities, today's top models still stumble over real physics.** As shown in Tab. 7, the best-performing model, o3 (high), achieves an accuracy of 58.90%, followed by DeepSeek-R1 at 55.30%, while most other models remain below 50%. In contrast, on challenging math tasks like AIME2025, o3 (high) reaches nearly 90% accuracy [41], and DeepSeek-R1 achieves 65% [13]. This indicates that although current LLMs demonstrate strong mathematical reasoning abilities, they still face significant challenges in complex physics reasoning.

**Models exhibit varying strengths across different physics subjects.** As shown in the Subject column of Tab. 7 and Fig. 5, models exhibit similar domain-specific patterns: poor performance on Thermodynamics, but better results on Mechanics and Electromagnetism, likely due to (1) inherent difficulty differences, with Thermodynamics being a more advanced and specialized topic, while the others are more broadly covered in a variety of datasets. (2) Imbalanced pre-training data distribution, as pre-training datasets may contain more samples from certain physics domains, significantly influencing the performance of models in those areas and leading to discrepancies in their overall capabilities.

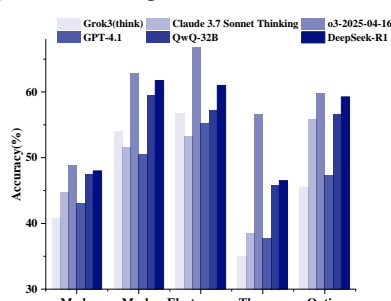

Figure 4: Performance of the (partial) models across different physics subjects.

Table 7: Main results on our **PHYSICS** test set evaluated by accuracy(%). **Rule Acc** and **Hybrid Acc** denote accuracies under rule-based and rule+model evaluation protocols. Subjects are divided **Mod.** (Modern Physics), **Mech.** (Mechanics), **Electromag.**(Electromagnetism), **Thermo.** (Thermodynamics), and **Optics**. Difficulty levels include: 1: High School and Below, 2: High School Olympiad, 3: Undergraduate (Non-Physics Major), 4: Undergraduate/Postgraduate(Physics Major). **EN** and **ZH** indicate English and Chinese inputs. **Bold** values mark the best overall per column; Underlined values highlight the best within each model category.

| Model | Accuracy | | Subject | | | | | Difficulty Level | | | | Language | |
|---|---|---|---|---|---|---|---|---|---|---|---|---|---|
| | Rule Acc | Hybrid Acc | Mod. | Mech. | Electromag. | Thermo. | Optics | 1 | 2 | 3 | 4 | EN | ZH |
| *Closed-source Reasoning Models* | | | | | | | | | | | | | |
| Gemini 2.5 Pro-0325 [16] | 23.45 | 45.05 | 37.25 | 49.00 | 51.75 | 35.50 | 51.75 | 77.08 | 43.37 | 63.16 | 38.49 | 44.70 | 45.20 |
| Grok 3 (Think) [59] | 25.10 | 46.40 | 40.75 | 54.00 | 56.75 | 35.00 | 45.50 | 56.25 | 54.79 | 55.24 | 42.28 | 47.90 | 44.90 |
| Claude 3.7 Sonnet Thinking [2] | 21.60 | 48.75 | 44.75 | 51.50 | 53.25 | 38.50 | 55.75 | 68.75 | 53.57 | 58.85 | 44.17 | 48.70 | 48.80 |
| o3 (high) [41] | 23.30 | **58.90** | **48.75** | **62.75** | **66.75** | **56.50** | **59.75** | **87.50** | **71.81** | **73.10** | **52.06** | **58.70** | **59.10** |
| *Closed-source Chat Models* | | | | | | | | | | | | | |
| Claude 3.7 Sonnet [2] | 21.45 | 44.15 | 42.25 | 45.00 | 50.75 | 31.50 | 51.25 | 72.92 | 52.55 | 58.85 | 37.29 | 44.10 | 44.20 |
| GPT-4.1 [40] | 21.30 | 46.75 | 43.00 | 50.50 | 55.25 | 37.75 | 47.25 | 87.50 | 60.64 | 55.95 | 41.03 | 45.90 | 47.60 |
| *Open-source Reasoning Models* | | | | | | | | | | | | | |
| DeepSeek-R1-Distill-Qwen-7B [13] | 13.80 | 35.65 | 32.25 | 44.00 | 40.75 | 22.25 | 39.00 | 66.67 | 43.88 | 51.20 | 28.48 | 37.90 | 33.40 |
| DeepSeek-R1-Distill-Llama-8B [13] | 8.80 | 22.70 | 21.25 | 28.25 | 22.50 | 16.50 | 25.00 | 45.83 | 29.59 | 34.21 | 17.26 | 27.50 | 17.90 |
| Qwen3-8B [46] | 21.50 | 45.65 | 41.25 | 52.00 | 50.75 | 35.25 | 49.00 | 79.17 | 52.04 | 60.05 | 39.01 | 48.40 | 42.90 |
| DeepSeek-R1-Distill-Qwen-32B [13] | 20.30 | 46.40 | 42.50 | 52.00 | 49.75 | 38.25 | 49.50 | 77.08 | 54.59 | 62.20 | 39.16 | 46.70 | 46.10 |
| QwQ-32B [51] | 20.65 | 53.30 | 47.50 | 59.50 | 57.25 | 45.75 | 56.50 | 85.42 | 56.12 | 68.18 | 47.09 | 53.00 | 53.60 |
| Qwen3-32B [46] | 21.10 | 47.25 | 46.00 | 51.25 | 51.25 | 40.25 | 47.50 | 81.25 | 50.51 | 58.61 | 42.00 | 49.40 | 45.10 |
| DeepSeek-R1-Distill-LLaMa-70B [13] | 19.55 | 45.50 | 40.75 | 51.25 | 48.25 | 36.00 | 51.25 | 75.00 | 52.55 | 59.09 | 39.16 | 45.90 | 45.10 |
| DeepSeek-R1 [13] | 27.55 | 55.30 | 48.00 | 61.75 | 61.00 | 46.50 | 59.25 | 83.33 | 59.69 | 67.46 | 49.85 | 53.50 | 57.40 |
| *Open-source Chat Models* | | | | | | | | | | | | | |
| Mistral-Nemo-Instruct-2407 [38] | 1.20 | 13.00 | 14.71 | 14.41 | 15.35 | 6.77 | 13.43 | 23.53 | 9.52 | 22.67 | 8.01 | 14.20 | 11.60 |
| Qwen2.5-7B-Instruct [45] | 6.95 | 22.30 | 22.50 | 25.00 | 28.00 | 12.75 | 23.25 | 50.00 | 29.59 | 35.17 | 16.22 | 24.30 | 20.30 |
| LLaMA3.1-8B-Instruct [17] | 3.35 | 13.10 | 14.25 | 13.00 | 18.50 | 7.25 | 12.50 | 27.08 | 18.37 | 20.81 | 9.42 | 14.20 | 12.00 |
| Gemma2-9B [50] | 2.25 | 16.20 | 14.00 | 17.25 | 22.00 | 8.25 | 19.50 | 35.42 | 23.98 | 25.36 | 11.51 | 17.00 | 15.40 |
| DeepSeek-MOE-16B-Chat [12] | 2.40 | 6.00 | 8.50 | 3.25 | 8.75 | 3.25 | 4.00 | 25.00 | 9.18 | 6.46 | 4.71 | 6.70 | 5.30 |
| LLaMA3.3-70B-Instruct [17] | 16.50 | 30.40 | 28.75 | 31.25 | 36.50 | 22.25 | 33.25 | 72.92 | 36.73 | 40.91 | 24.66 | 30.60 | 30.20 |
| Qwen2.5-72B-Instruct [45] | 17.00 | 32.25 | 30.00 | 35.50 | 38.50 | 21.00 | 36.25 | 62.50 | 38.78 | 45.45 | 26.08 | 33.00 | 31.50 |
| Mistral-Large-Instruct-2407 [37] | 17.50 | 35.10 | 36.25 | 39.25 | 36.75 | 23.50 | 39.75 | 70.83 | 37.24 | 47.37 | 29.67 | 35.40 | 34.80 |
| DeepSeek-V3 [13] | 22.45 | 47.05 | 47.00 | 51.75 | 49.25 | 35.25 | 51.75 | 79.17 | 51.02 | 58.37 | 41.70 | 46.70 | 47.40 |

**Open-source models are gradually catching up with closed-source models, but a performance gap remains.** The current strongest open-source model, DeepSeek-R1, has already surpassed many closed-source models in physics reasoning, including strong ones like Gemini-2.5-Pro. Other open-source models like Qwen3-8B also show strong performance. However, closed-source models still maintain the overall lead, with OpenAI-o3 continuing to outperform all open-source models, highlighting a performance gap that warrants further investigation.

**Reasoning models clearly outperform chat models.** As expected, reasoning models excel in solving complex physics problems compared to general chat models. Whether comparing o3 with GPT-4.1 among closed-source models or DeepSeek-R1 with DeepSeek-V3 among open-source models, a clear trend emerges: reasoning models significantly outperform non-reasoning models within the same generation. This highlights the importance of developing specialized reasoning capabilities.

**Model accuracy is not fully correlated with the knowledge demand of the task.** As shown in the Difficulty Level column of Tab. 7, model accuracy declines with increasing problem difficulty. Surprisingly, models perform worse on high school Olympiad problems than on undergraduate non-physics-major questions. We believe this is because, although undergraduate problems require broader knowledge, models already possess some foundational physics knowledge. In contrast, high school Olympiad problems demand strong reasoning skills, which current models largely lack.

**The language of the question affects model performance.** As shown in the Language column of Tab. 7, models exhibit varying performance across languages. Since each question has an equivalent version in the other language, differences in accuracy suggest that LLMs process languages differently. Smaller models tend to show larger gaps. For example, DeepSeek-R1-Distill-LLaMa-7B shows a 10% difference between English and Chinese, likely due to limited model capacity.

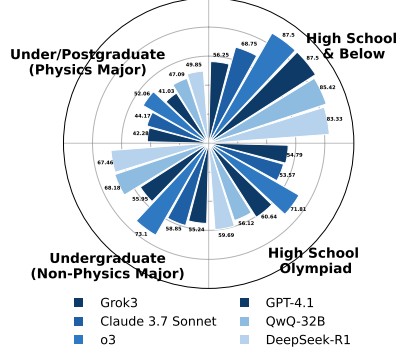

Figure 5: Performance of the (partial) model across different physics difficulties.

Table 8: Evaluation results using our Rule+Model method after post-training on our training dataset.

| Model | SFT | Physics | | | | Math | |
| | | PHYSICS (ours) | GPQA (Physics) | OlympiadBench (Physics) | UGPhysics | MATH-500 | AIME-2025 |
|---|---|---|---|---|---|---|---|
| Qwen2.5-3B-Instruct | ✗ | 13.65 | 28.19 | 15.95 | 14.34 | 62.40 | 0.00 |
| | ✓ | 19.00 (↑ **5.35**) | 31.27 (↑ **3.08**) | 19.94 (↑ **3.99**) | 19.18 (↑ **4.84**) | 63.20 (↑ **0.90**) | 6.67 (↑ **6.67**) |
| Qwen2.5-7B-Instruct | ✗ | 22.30 | 35.68 | 23.36 | 19.67 | 76.00 | 0.00 |
| | ✓ | 32.85 (↑ **10.55**) | 42.73 (↑ **7.05**) | 31.90 (↑ **8.54**) | 27.74 (↑ **8.08**) | 81.60 (↑ **5.60**) | 13.33 (↑ **13.33**) |
| Qwen2.5-14B-Instruct | ✗ | 27.65 | 44.93 | 29.34 | 26.53 | 80.16 | 0.00 |
| | ✓ | 40.20 (↑ **12.55**) | 60.35 (↑ **15.42**) | 45.86 (↑ **16.52**) | 41.92 (↑ **15.39**) | 89.60 (↑ **9.44**) | 22.50 (↑ **22.50**) |
| Llama3.2-3B-Instruct | ✗ | 8.18 | 23.84 | 5.98 | 7.29 | 38.12 | 0.83 |
| | ✓ | 18.44 (↑ **10.26**) | 31.94 (↑ **8.10**) | 16.24 (↑ **10.26**) | 19.73 (↑ **12.44**) | 47.23 (↑ **9.11**) | 1.25 (↑ **0.42**) |
| Llama3.1-8B-Instruct | ✗ | 12.36 | 24.45 | 7.41 | 12.59 | 45.67 | 0.42 |
| | ✓ | 21.94 (↑ **9.58**) | 34.86 (↑ **10.41**) | 16.95 (↑ **9.54**) | 21.68 (↑ **9.09**) | 49.03 (↑ **3.36**) | 2.08 (↑ **1.66**) |
| Mistral7B-Instruct-v0.3 | ✗ | 7.84 | 18.39 | 5.98 | 10.24 | 15.25 | 0.00 |
| | ✓ | 10.12 (↑ **2.28**) | 20.21 (↑ **1.39**) | 8.26 (↑ **2.28**) | 12.04 (↑ **1.77**) | 18.50 (↑ **3.25**) | 0.42 (↑ **0.42**) |
| Mistral8B-Instruct-2410 | ✗ | 13.74 | 28.30 | 11.97 | 15.20 | 54.03 | 2.50 |
| | ✓ | 17.93 (↑ **4.19**) | 32.87 (↑ **4.57**) | 15.95 (↑ **3.98**) | 18.92 (↑ **3.72**) | 58.48 (↑ **4.45**) | 3.13 (↑ **0.63**) |

Table 9: Evaluation results using our SFT and other reasoning enhancement methods.

| Model | Physics | | | | Math | |
| | PHYSICS (ours) | GPQA (Physics) | OlympiadBench (Physics) | UGPhysics | MATH-500 | AIME-2025 |
|---|---|---|---|---|---|---|
| Qwen2.5-7B-base | 7.86 | 11.73 | 7.12 | 7.44 | 42.83 | 1.67 |
| SimpleRL-Qwen2.5-7B [64] | 26.49 | _38.77_ | 22.22 | 24.20 | **77.60** | _10.42_ |
| General-Reasoner-Qwen2.5-7B [36] | _27.66_ | 35.19 | _26.78_ | **26.88** | 77.50 | 5.83 |
| Absolute-Zero-7B [66] | 19.86 | 28.03 | 14.81 | 18.08 | 72.50 | 10.20 |
| PHYSICS-7B-base (ours) | **32.67** | **47.80** | **29.34** | _26.38_ | _77.45_ | **11.67** |
| Qwen2.5-14B-base | 14.53 | 19.71 | 12.82 | 9.08 | 55.75 | 3.33 |
| SimpleRL-Qwen2.5-14B [64] | _31.77_ | 44.77 | 30.20 | 28.80 | _81.50_ | 13.75 |
| General-Reasoner-Qwen2.5-14B [36] | 30.39 | _46.37_ | _34.76_ | _33.28_ | 80.75 | _16.25_ |
| Absolute-Zero-14B [66] | 24.94 | 42.40 | 29.34 | 25.68 | 78.87 | 12.50 |
| PHYSICS-14B-base (ours) | **35.13** | **52.28** | **35.62** | **37.88** | **83.22** | **18.33** |

**Larger models from the same family generally outperform smaller ones.** Whether looking at Qwen or LLaMA series models, we observe consistent improvements in performance as model scale increases. This aligns with common expectations and supports the validity of the scaling law.

**Hybrid verification leads to a clear accuracy boost.** Across all model classes, the hybrid verifier achieves greatly higher accuracy than the rule-based verifier alone. The gap often exceeds 20%, highlighting the importance of combining rules and models in evaluating physics questions.

## 4.2 Training Result and Analysis

For training, we select models with various parameter sizes, including: Qwen2.5-3B-Instruct, Qwen2.5-7B-Instruct, Qwen2.5-14B-Instruct [45], Llama3.2-3B-Instruct, Llama3.1-8B-Instruct [17], Mistral7B-Instruct-v0.3, Mistral8B-Instruct-2410 [37]. These models are fine-tuned using our training dataset to enhance the models' reasoning capabilities in physics problem-solving. Here, we use QwQ-32B to generate detailed reasoning paths for 4,000 samples from the training set, which are then used for training. The goal is to enable weaker models to learn basic physical reasoning abilities from this data. The configuration of the supervised fine-tuning (SFT) can be found in Appendix E.

Table 8 presents the performance of LLMs on physics and mathematics benchmarks following SFT on our physics-focused training dataset. The results show that fine-tuned models show notable improvements in both physics and mathematics. In the following, we analyze the training results from multiple perspectives to illustrate the validity and effectiveness of the training data.

**Improved Physics Performance Across Diverse Benchmarks.** The physics training dataset consistently enhances model performance across a variety of physics benchmarks, including olympiad-level and undergraduate-level problems. Here, we conduct evaluations on GPQA [47], Olympiad-Bench [19], UGPhysics [61], and our PHYSICS test set. Fine-tuned models show significant improvements, reflecting the dataset's comprehensive coverage of physics sub-disciplines and difficulty levels, which strengthens the models' ability to tackle diverse physical problem-solving tasks.

Meanwhile, in Tab. 9, we used the Qwen series models as the base model and compared our SFT with several enhanced reasoning methods. Our trained models outperformed others on most physics datasets, showing that our data boosts both reasoning and physics knowledge, with slight improvements in math reasoning indicating skill transfer. This further verifies that our dataset not only enhances the model's reasoning ability but also strengthens its understanding of physics.

**Math Performance Gains from Physics Training.** We evaluate the mathematical capabilities of the trained model on MATH500 [22] and AIME-2025 [42]. Due to the small number of problems in AIME-2025 causing large fluctuations, we report the average results over 16 runs. The physics-based training also contributes to improvements in mathematics abilities, particularly in advanced problems such as AIME-2025, indicating that skills learned in physics contexts can transfer to mathematical reasoning. This suggests that physics and mathematics can mutually enhance each other.

**General Domain Performance Gains from Physics Training.** To verify whether our training improved the model's general reasoning ability, we compare the performance of several Qwen-2.5 models on GPQA-Diamond [47] and MMLU-Pro [58] before and after training. These are two benchmarks that focus on assessing the general-domain capabilities of large language models (LLMs), testing their ability to reason across a wide range of topics. Results presented in Tab. 10 show that all 3B, 7B, and 14B parameter models improved after fine-tuning with our data, demonstrating its effectiveness in enhancing general reasoning skills. We also observed that larger models tended to show greater improvements, likely due to their higher capacity for learning complex patterns and generalizing from diverse data.

Table 10: Evaluation results after post-training.

| Model | GPQA-D | MMLU-pro |
|---|---|---|
| Qwen2.5-3B-Instruct | 30.37 | 39.39 |
| +SFT | **30.89** | **39.95** |
| Qwen2.5-7B-Instruct | 33.46 | 52.68 |
| +SFT | **39.08** | **54.11** |
| Qwen2.5-14B-Instruct | 39.65 | 58.30 |
| +SFT | **49.62** | **65.77** |

## 5 Error Analysis

In this section, we present a detailed analysis of the errors observed during evaluation. We categorize the mistakes made by models in physics reasoning into two types: Knowledge Deficit and Reasoning Flaw. To illustrate these categories, we examine the reasoning process of LLMs, particularly Grok 3 (Think). Next, we first introduce the two types of errors in Sec. 5.1, and then provide an in-depth analysis in Sec. 5.2, highlighting the limitations of large models in solving physics problems.

### 5.1 Understand Reasoning Errors

During our manual inspection of the reasoning processes of multiple models, we find that existing model errors can be categorized into two main aspects:

**Knowledge Deficit.** This type of error primarily refers to mistakes caused by the model's incorrect understanding or application of physics knowledge, as shown in Fig. 6 (left). We further divide it into two categories: **Conceptual Errors** and **Modeling Errors**. Conceptual Errors occur when the model fails to correctly understand or apply fundamental physics concepts. For example, misunderstanding the concept of acceleration may lead to an incorrect solution. Modeling Errors refer to issues such as mixing ideal and non-ideal boundary conditions, ignoring or omitting constraints. For instance, a model might directly ignore friction when it should be considered. Addressing these errors requires better comprehension of physical principles and domain knowledge.

**Reasoning Flaw.** This category includes errors that occur during the reasoning process, as illustrated in Fig. 6 (right). We classify them into two types: **Comprehension Misunderstanding** and **Computational Errors**. Comprehension Misunderstanding arises when the model misinterprets the question or its context, leading to deviations from the correct reasoning path and ultimately resulting in incorrect conclusions. Computational Errors refer to mistakes made during mathematical or logical computation, such as arithmetic miscalculations or incorrect application of formulas. Resolving this type of error requires improvements in both language understanding and reasoning capabilities, as well as refining the model's ability to handle complex operations.

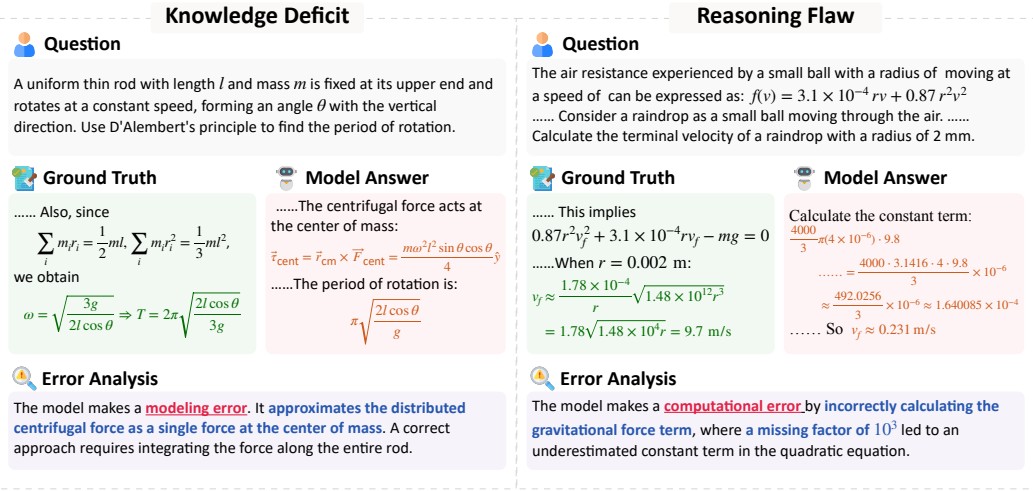

Figure 6: **Representative error cases of Knowledge Deficit (left) and Reasoning Flaw (right).** The model answers are generated by Grok 3 (Think).

## 5.2 Analyze Reasoning Errors

First, we analyze the distribution of different types of errors made by the model. We selected 100 incorrect answers generated by Grok 3 (Think) and used human experts to annotate the error categories. The distribution of errors is shown in Fig. 7.

As can be seen, Knowledge Deficit accounts for a large portion of the errors, with Conceptual Errors and Modeling Errors together contributing nearly 60% of all error cases. This suggests that **even the most advanced models still lack sufficient physics knowledge**, leading to confusion in applying physical principles during reasoning. To address these errors, we propose that improvements should start from the model's training stage, enhancing its understanding and application of physics principles and knowledge. This will require specialized design for the physics domain. At the same time, **reasoning flaws cannot be ignored**. For this type of error, we believe models require not only better consistency in long-chain reasoning processes but also stronger foundational reasoning capabilities. Advances in this area could benefit from interdisciplinary development, particularly in conjunction with fields such as mathematics.

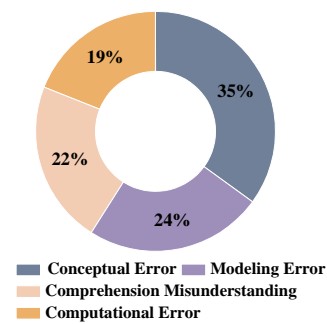

Figure 7: Proportion of error types by the reasoning model.

In summary, our error analysis reveals that physics reasoning, compared to mathematical deduction, demands additional domain-specific knowledge and involves real-world implications, making it inherently more complex. These findings highlight important challenges for the continued advancement of language models in handling structured and grounded reasoning tasks.

## 6 Conclusion

In this work, we introduce a large-scale, high-quality physics dataset with a wide range of difficulty levels. We also provide a clear split into training and test sets, to support both the improvement and evaluation of models' physics reasoning abilities. Due to the bias of current evaluation frameworks in the physics domain, we design a specialized evaluation method tailored to physics problems. Our experimental results show that even state-of-the-art LLMs have limited performance on our PHYSICS. At the same time, fine-tuning with high-quality PHYSICS data proves to be effective in enhancing model capabilities in this domain. This highlights the current limitations of LLMs in physics reasoning, while also pointing to new challenges and opportunities for future development. We believe our work can contribute to the advancement of Large Language Models in general.

## Acknowledgments

This work was supported by the Shanghai Artificial Intelligence Laboratory and a locally commissioned task from the Shanghai Municipal Government.

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

# Appendix for PHYSICS

## A    Statistics of PHYSICS

The following part displays the Chinese and English bilingual versions of the same question in PHYSICS.

---

*A Chinese Question of PHYSICS*

**id:** 3543
**question:** 早在20世纪20年代，Ramsauer和Townsend各自独立地发现对于能量约0.4eV的电子，在气态氩原子上的散射截面比几何散射截面（$\pi a^2$，其中 $a$ 为Atomic radius）小得多。问反常散射截面的起源为何？
**solution:** 当吸引势足够强时，在某一能量处 $l = 0$ 的分波可能被拉入半周，其相移 $\delta_0$ 为 $\pi$。此时 $l = 0$ 的分波对散射截面没有贡献，而其他分波的贡献又很小（能量很低），因而散射截面变得很小，这就是所谓的 Ramsauer-Townsend 效应。
**answer:** $\boxed{\delta_0 = \pi}$
**answer_type:** ["Numerical"]
**language:** zh
**domain:** Advanced Physics
**difficulty:** Physics UnderGraduate
**translate:** false

---

*An English Question of PHYSICS*

**id:** 3543
**question:** As early as the 1920s, Ramsauer and Townsend independently discovered that for electrons with an energy of approximately 0.4 eV, the scattering cross-section on gaseous argon atoms is much smaller than the geometric scattering cross-section ($\pi a^2$, where $a$ is the atomic radius). What is the origin of this anomalous scattering cross-section?
**solution:** When the attractive potential is strong enough, the partial wave with $l = 0$ may be drawn into a half-cycle at a certain energy, and its phase shift $\delta_0$ becomes $\pi$. At this point, the partial wave with $l = 0$ does not contribute to the scattering cross-section, and the contributions from other partial waves are very small (very low energy), resulting in a very small scattering cross-section. This is known as the Ramsauer-Townsend effect.
**answer:** \boxed{$\delta_0 = \pi$}
**answer_type:** ["Numerical"]
**language:** en
**domain:** Advanced Physics
**difficulty:** Physics UnderGraduate
**translate:** true

---

# B Details of PHYSICS pipeline

When collecting physics data, we designed the processing pipeline illustrated in Fig. 8. First, raw PDF physics textbooks are converted into Markdown format. Next, a large language model (LLM) is employed to extract question-and-answer pairs, individual questions, and individual answers from the converted text. To minimize potential hallucinations by the LLM, the extracted content is cross-verified against the original source text. Finally, we analyze the structural features of each text fragment to accurately align individual questions with their corresponding answers.

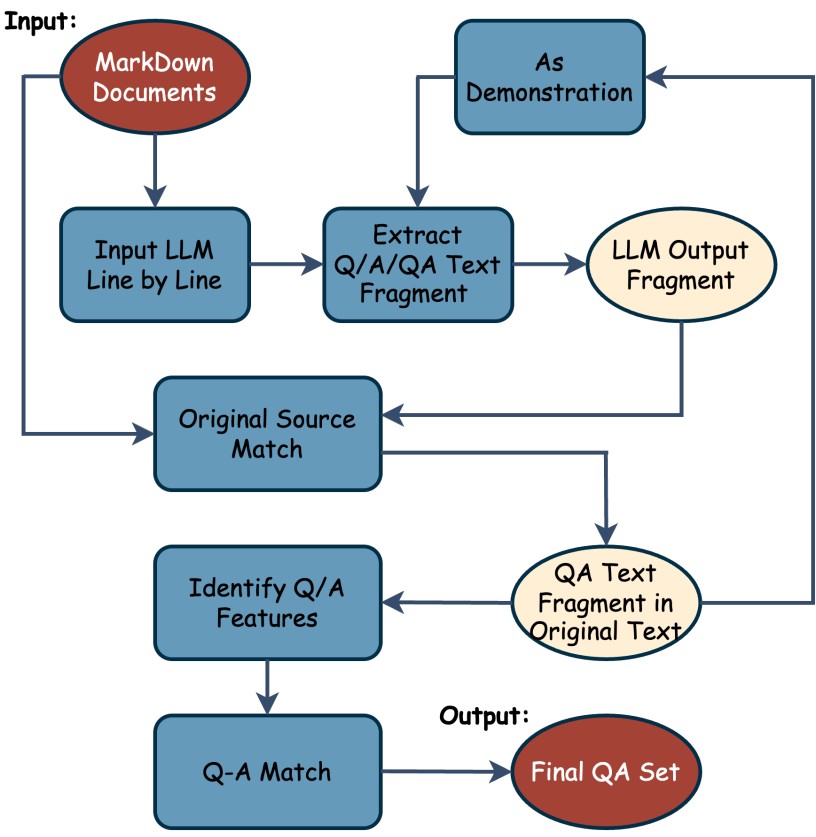

Figure 8: Pipeline of PHYSICS Data Processing.

---

*Prompt of QA Pair Extraction*

Extract the first complete question (Type:Q) or the first answer (Type:A) or the first question-answer pair (Type:QA) content from the following input text. Only include the first Q/A/QA that appears. It is necessary to make sure the exercise is complete. If there is nothing to output, return "None".
Tips:
1. A question or answer usually begins with headings numbered as "1.", "1.1", "1-1", and so on.
2. Sub-questions under the same main question should be grouped within the same exercise.
3. The answer related to the question usually begins with "Solution", and so on.
4. The answer might be very concise, without any derivation process.
**Output Format:**
**Type:**
**Content:**

Now try to extract the exercise from the following input text.
**Input:**

**Output:**

For translation, we used GPT-4o for mutual translation. The prompt is shown below:

---

***Prompt of Translation***

Please act as an expert in physics and translation. Your task is to translate the Chinese/English text I provide into English/Chinese.
The text will be a physics question or an answer, and your translation should reflect it accurately.
If there are LaTeX expressions, numbers, or units, do not translate them.
Adhere strictly to the meaning of the original text. Do not add or remove any content.
Only output the translated version of the original text.
The content you need to translate is: {{text}}
Translation:

---

We ensured translation quality through LLM and human evaluations. Gemini-2.5-Flash first assessed each translation using GEMBA-MQM [28], focusing on accuracy, fluency, style, and terminology; low-quality outputs were retranslated. Then, over five experts reviewed them using MQM [33], with manual corrections. This yielded high-quality translations, doubling the dataset to 16,568 samples.

Regarding the training data, we currently use QwQ-32B to perform eight rounds of rejection sampling on the training set, generating 4,000 samples with detailed and accurate reasoning paths. The remaining data either contain brief reasoning traces extracted from data sources or only provide final answers. Since the goal of the training section is to demonstrate the effectiveness of our dataset in enhancing models' physical reasoning capabilities, we currently utilize only the data with detailed reasoning paths for SFT. In future releases, we plan to include more detailed reasoning paths generated by stronger models (e.g., DeepSeek-R1, Qwen3-235B-A22B) to further support the community.

Table 11: Inference Configuration Parameters

| Parameter | Value |
|---|---|
| *Model and Engine Configuration* | |
| Enable Chunked Prefill | True |
| Enable Prefix Caching | True |
| *Sampling Parameters* | |
| Top-p | 0.95 |
| Top-k | 20 |
| Temperature | 0.6 |
| Maximum Tokens | 16384 |
| Repetition Penalty | 1.1 |
| Seed | 42 |

## C  Details of Evaluation

Table 11 shows the configuration of inference. All local inference is run on a server equipped with eight NVIDIA A800 GPUs.

We used the following prompt for inference to generate model output in preparation for evaluation.

---

***Prompt of Inference***

Below is an open-ended problem in undergraduate-level Physics. Please answer this problem adhering to the following rules:
1. Please use LaTeX format to represent the variables and formulas used in the solution process and results.
2. Please put the final answer(s) in \boxed{}, note that the unit of the answer should not be included in \boxed{}.
3. If there are multiple final answers, please separate them by commas in \boxed{},
e.g., \boxed{answer 1, answer 2}.
Problem: {prompt}

---

# D   Details of Evaluation Framework

During the evaluation process, we used the following prompt to judge the model results.

---

**Prompt of Evaluation**

You are a diligent and precise assistant tasked with evaluating the correctness of responses. You will receive a question, an output sentence, and the correct answer. Your task is to determine if the output sentence accurately answers the question based on the provided correct answer. Respond with either [Correct] or [Incorrect].
Special considerations:
1. **Multiple Answers**: If the output contains multiple answers, evaluate whether later answers modify or correct earlier ones. In such cases, compare the final answer with the correct answer. If the final answer is unclear or incorrect, respond with [Incorrect].
2. **Mathematical Problems**: If the formats differ but the answers are mathematically equivalent, respond with [Correct].
3. **Explicit Options**: If the question provides explicit candidate answers, the output will be considered correct if it clearly indicates the correct option's code or the correct option's content.
4. **No Explicit Options**: If the question does not provide explicit options, the output must align with the correct answer in content and meaning to be considered [Correct].
Question: {problem}, Output sentence: {given_answer}, Correct answer: {ground_truth}, Judgement:

---

## D.1   Training Set and Test Set

For physics problems, we construct equivalent versions of questions with the same answers from GPQA-Physics to build our training and test sets. To ensure that the evaluated models also demonstrate generalization capability on physics-related tasks, we further include mathematical problems from MATH-500. The detailed construction process is as follows.

To prevent data leakage, we select some data from the high school difficulty level of our own test set to build the test set for xVerify-Physics. First, we use GPT-4o to filter questions in the dataset that match the error cases. GPT-4o then distills incorrect answers based on the original answer and equivalent correct answers, which undergo human review. We also include math questions, using MATH-500 to distill incorrect and correct answers in the same manner. Note that we do not use the original questions during application, only the distilled answers. And a single question can be distilled into five questions.

Table 12: Training and Test Set Data Source.

| Category | Training | Test |
|---|---|---|
| GPQA-Physics | 60 | 0 |
| PHYSICS-Highschool | 0 | 135 |
| MATH-500 | 45 | 120 |

Table 13: Training and Test Set Construction.

| Category | Training Set | Test Set |
|---|---|---|
| Physics Incorrect | 274 | 544 |
| Physics Correct | 238 | 624 |
| Math Incorrect | 184 | 522 |
| Math Correct | 222 | 548 |
| **Overall** | 918 | 2238 |

We provide an overview of the data sources and construction of the training and test sets for training xVerify-Physics. Tab. 12 details the data sources for the training and test sets across different categories. We sampled 60 questions from GPQA-Physics and 45 questions from MATH-500 as the basis for the training set. For the test set, we selected 135 questions from PHYSICS-HighSchool and 120 questions from MATH-500. Tab. 13 summarizes the construction of the training and test sets, classified by correct and incorrect responses to physics and math problems. *Incorrect* and *Correct* refer to the equivalence matching settings in the process of constructing equivalent answers for the questions. Specifically, we set the answer pairs that need to be judged as not equivalent (Incorrect) or equivalent (Correct) during the matching process.

## D.2 Prompt for Data Distilling

We used GPT-4o for data distillation. By inputting questions and their original answers, we required the model to output answers aligned with our error cases, specifically related to unit conversion and numerical simplification. Two versions of prompts were used to generate correct answers and incorrect answers, respectively.

---

**Generate Correct Answers**

You are required to provide different equivalent forms of the standard answer for the following physics problem, focusing on fraction simplification and unit conversion to express the answer in varied representations.

**Problem**: [Insert physics problem here]

**Answer**: [Insert standard answer here]

**Example 1:**

**Problem**: In an experiment, the wavelength of light is measured using single-slit diffraction. The slit width is $a = 0.2\,\text{mm}$, the distance from the slit to the screen is $L = 1.5\,\text{m}$, and the distance from the first dark fringe to the central bright fringe on the screen is $x = 4.5\,\text{mm}$. Find the wavelength of light $\lambda$.

**Answer**: $\frac{3}{5} \times 10^{-6}$

**Output**:

```
{
    "answer1": "600 \\text{nm}",
    "answer2": "600",
    "answer3": "0.6",
    "answer4": "6 \\times 10^{-5} \\text{cm}",
    "answer5": "6 \\times 10^{-4} \\text{mm}"
}
```

**Example 2:**

**Problem**: While studying thermodynamics, a certain ideal gas expands from a volume of $V_1 = 2\,\text{L}$ to $V_2 = 6\,\text{L}$ during an isothermal process, with the work done by the surroundings on the gas being $W = 1000\,\text{J}$. Given that the gas temperature remains constant, find the gas pressure $P$ (based on the initial pressure, assuming the relationship between initial pressure and volume satisfies the ideal gas equation of state).

**Answer**: $\frac{10^6}{2\ln 3}$

**Output**:

```
{
    "answer1": "455.1",
    "answer2": "\\frac{5 \\times 10^{5}}{\\ln 3}",
    "answer3": "455199.4",
    "answer4": "455.1 \\text{kPa}"
}
```

**Please note:**

1. You need to provide 3 to 6 different standard forms of the answer.

2. Each different form must be equivalent to the standard answer, i.e., it should still be a correct and valid answer.

3. You may use LaTeX, scientific notation, or other standard mathematical expressions.

4. Please follow the JSON format below for the output:

```
{
    "answer1": "xxx",
    "answer2": "xxx",
    "answer3": "xxx",
    ...
}
```

You are required to provide different forms of the standard answer for the following physics problem, focusing on fraction simplification and unit conversion. However, the answers must be completely incorrect in terms of fraction simplification and unit conversion, resulting in values and units that are entirely wrong and not equivalent to the standard answer.

**Problem**: problem

**Answer**: answer

**Example 1:**

**Problem**: In an experiment, the wavelength of light is measured using single-slit diffraction. The slit width is $a = 0.2\,\text{mm}$, the distance from the slit to the screen is $L = 1.5\,\text{m}$, and the distance from the first dark fringe to the central bright fringe on the screen is $x = 4.5\,\text{mm}$. Find the wavelength of light $\lambda$.

**Answer**: $\frac{3}{5} \times 10^{-6}$

**Output**:

```
{
    "answer1": "12 \\, \\text{kg}",
    "answer2": "0.3 \\, \\text{J}",
    "answer3": "45 \\, \\text{s}",
    "answer4": "7",
    "answer5": "9 \\times 10^2 \\, \\text{Pa}",
    "answer6": "0.002 \\, \\text{mol}"
}
```

**Example 2:**

**Problem**: While studying thermodynamics, a certain ideal gas expands from a volume of $V_1 = 2\,\text{L}$ to $V_2 = 6\,\text{L}$ during an isothermal process, with the work done by the surroundings on the gas being $W = 1000\,\text{J}$. Given that the gas temperature remains constant, find the gas pressure $P$ (based on the initial pressure, assuming the relationship between initial pressure and volume satisfies the ideal gas equation of state).

**Answer**: $\frac{10^6}{2\ln 3}$

**Output**:

```
{
    "answer1": "500 \\, \\text{m}",
    "answer2": "2 \\times 10^{-3} \\, \\text{kg}",
    "answer3": "1000 \\, \\text{s}",
    "answer4": "0.8 \\, \\text{mol}",
    "answer5": "300 \\, \\text{W}"
}
```

**Please note:**

1. You need to provide 3 to 6 different standard forms of the answer.

2. Each form must be completely incorrect, using wrong units and values that are not equivalent to the standard answer in any way.

3. You may use LaTeX, scientific notation, or other standard mathematical expressions.

4. Please follow the JSON format below for the output:

```
{
    "answer1": "xxx",
    "answer2": "xxx",
    "answer3": "xxx",
    ...
}
```

## D.3 Physics-xVerify Training Seting

Table 14 shows the parameters used during SFT. We choose xVerify-8B-I as the base model.

Table 14: Experimental Parameters for xVerify-Physics SFT

| Parameter | Value |
|---|---|
| *Model Arguments* | |
| Model Name | xVerify-8B-I |
| Attention Implementation | flash_attention_2 |
| *SFT Trainer Configuration* | |
| Use LoRA | True |
| LoRA Target Modules | all-linear |
| LoRA Rank | 16 |
| LoRA Alpha | 32 |
| LoRA Dropout | 0.05 |
| BF16 | True |
| Gradient Checkpointing | False |
| Learning Rate | $5 \times 10^{-5}$ |
| LR Scheduler Type | cosine_with_min_lr |
| Minimum LR Rate | 0.1 |
| Packing | False |
| Maximum Sequence Length | 1024 |
| Maximum Steps | -1 |
| Number of Training Epochs | 2 |
| Gradient Accumulation Steps | 4 |
| Per Device Train Batch Size | 2 |
| Per Device Eval Batch Size | 2 |
| GPUs Per Node | 2 |
| Number of Nodes | 1 |
| Seed | 42 |
| Use Liger Kernel | True |
| Warmup Ratio | 0.02 |

## D.4 Misjudgement Cases in Evaluation

For Case 1, it is the only one incorrect case misjudged as correct in section 3.3, with all others being correct.

For Case 2 and Case 3, our analysis found that these two question types are prone to misjudgement, which overlaps with the error-prone question types identified in Section 3.3 but also presents new forms. This indicates that addressing error-prone question types requires a more diverse training set to further improve evaluation accuracy.

---

**Case 1: *Only one incorrect instance was mistakenly judged as correct***

**Question:** The absorption coefficient of a uniform medium is $\alpha = 0.32\,\mathrm{cm}^{-1}$. Find the thickness of the medium when the transmitted light intensity is 0.5 times the incident light intensity.

**Ground Truth:** 2.1661, cm

**Model Output:** 1.5, A

**Human Annotation:** Incorrect

**xVerify-Physics:** Correct

---

**Case 2: *Multi-answer question + Numerical Simplification***

**Question:** The equilibrium positions of particles $A$ and $B$ in a uniform medium are on the $x$-axis, with coordinates $x_A = 0$ and $x_B = 16$ cm. A simple harmonic transverse wave propagates in the positive $x$-direction with a wave speed of $v = 20$ cm/s, wavelength greater than 20 cm, amplitude $y_0 = 1$ cm, and no attenuation during propagation. At $t = 0$, the displacements of $A$ and $B$ from their equilibrium positions are equal in magnitude and direction, but their directions of motion are opposite. Thereafter, every $\Delta t = 0.6$ s, their displacements from equilibrium are equal in magnitude and direction. It is known that at time $t_1$ ($t_1 > 0$), particle $A$ is at a wave crest. Find: (1) Starting from $t_1$, the minimum time required for particle $B$ to be at a wave crest; (2) The displacement of particle $B$ from its equilibrium position at time $t_1$.

**Ground Truth:** 0.8, s, -0.5, cm

**Model Output:** 4/5, -1/2

**Human Annotation:** Correct

**XVerify-Physics:** Incorrect

---

**Case 3: *Multi-choice question + Unit Conversion***

**Question:** The photon emitted from the transition between the two hyperfine energy levels of the ground state of a cesium atom has a stable frequency. The energy difference between the two levels used in a cesium atomic clock is on the order of $10^{-5}$ eV. The frequency of the photon emitted from the transition is on the order of (Planck's constant $h = 6.63 \times 10^{-34}$ J · s, elementary charge $e = 1.60 \times 10^{-19}$ C):
A. $10^3$ Hz B. $10^8$ Hz C. $10^9$ Hz D. $10^{12}$ Hz

**Ground Truth:** C

**Model Output:** 1,000,000 kHz

**Human Annotation:** Correct

**xVerify-Physics:** Incorrect

---

# E  Details of Training

Table 15 shows the parameters used during training with the PHYSICS training set. All models in the Qwen series use the same parameters, with Qwen2.5-7B-Instruct as an example here.

Table 15: Experimental Parameters for Qwen2.5-7B-Instruct Post Training

| Parameter | Value |
|---|---|
| *Model Arguments* | |
| Model Name | Qwen2.5-7B-Instruct |
| Attention Implementation | flash_attention_2 |
| *SFT Trainer Configuration* | |
| BF16 | True |
| Gradient Checkpointing | True |
| Learning Rate | $5 \times 10^{-5}$ |
| LR Scheduler Type | cosine_with_min_lr |
| Minimum LR Rate | 0.1 |
| Packing | True |
| Maximum Sequence Length | 16384 |
| Number of Training Epochs | 3 |
| Per Device Train Batch Size | 4 |
| Per Device Eval Batch Size | 4 |
| GPUS Per Node | 8 |
| Number of Nodes | 1 |
| Seed | 42 |
| Use Liger Kernel | True |
| Warmup Ratio | 0.02 |

# F Detailed Analysis

## F.1 Language Performance Analysis

Fig. 9 shows that most models perform similarly in English (EN) and Chinese (ZH), with minor variations. o3 (high) achieves the highest accuracies in both languages (58.70% EN, 59.10% ZH), closely followed by DeepSeek-R1 (53.50% EN, 57.40% ZH) and QwQ-32B (53.00% EN, 53.60% ZH), demonstrating their language-agnostic robustness. In contrast, DeepSeek-MOE-16B-Chat (6.70% EN, 5.30% ZH) and Mistral-Nemo-Instruct-2407 (14.20% EN, 11.60% ZH) perform poorly, with particularly low scores in Chinese, suggesting limited multilingual capability. Some models, like DeepSeek-R1-Distill-Llama-8B, show a notable gap (27.50% EN vs. 17.90% ZH), indicating potential weaknesses in processing Chinese inputs. Overall, the close alignment of EN and ZH accuracies for top models suggests that language does not significantly impact performance on physics tasks, likely due to the mathematical nature of the problems. However, weaker models exhibit slightly lower performance in Chinese, possibly due to training data imbalances or linguistic complexities.

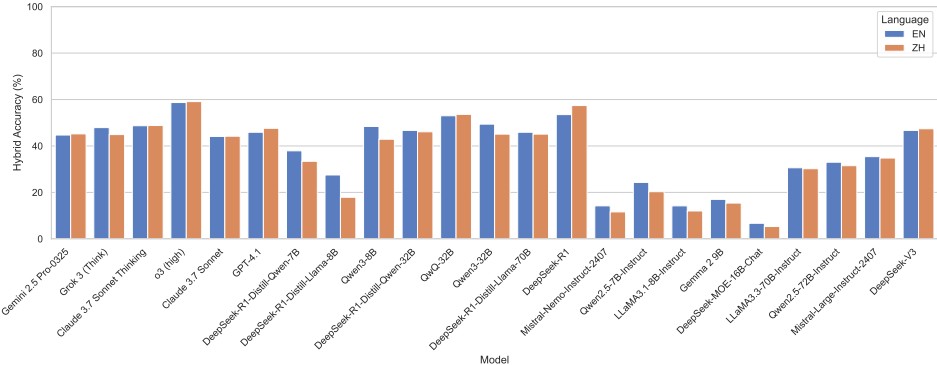

Figure 9: Model Performance by Language

## F.2 Subject Performance Analysis

Fig. 10 reveals significant variation in model capabilities across Modern Physics, Mechanics, Electromagnetism, Thermodynamics, and Optics. The model o3 (high) consistently achieves the highest Hybrid Accuracy across all subjects, with standout performances in Electromagnetism (66.75%), Mechanics (62.75%), and Optics (59.75%), indicating its robustness in handling diverse physics problems. DeepSeek-R1 and QwQ-32B also perform strongly, particularly in Mechanics (61.75% and 59.50%) and Electromagnetism (61.00% and 57.25%), positioning them as competitive open-source alternatives. Conversely, DeepSeek-MOE-16B-Chat and Mistral-Nemo-Instruct-2407 exhibit the lowest accuracies, with scores as low as 3.25% and 6.77% in Thermodynamics, highlighting their limitations in complex physics tasks. Thermodynamics appears to be the most challenging subject, with most models scoring lower (e.g., median around 35–40%) compared to Electromagnetism and Mechanics, where top models exceed 60%. This suggests that thermodynamics problems may require more specialized reasoning or knowledge that many models lack.

## F.3 Difficulty Level Performance Analysis

Fig. 11 illustrates a clear trend: model performance decreases as difficulty increases from High School (HS) to Undergraduate/Postgraduate Physics (UG/PG Phys). For HS-level problems, o3 (high) and GPT-4.1 tie for the highest accuracy at 87.50%, followed closely by QwQ-32B (85.42%) and DeepSeek-R1 (83.33%), indicating strong performance on foundational physics tasks. However, at the UG/PG Phys level, accuracies drop significantly, with o3 (high) leading at 52.06%, followed by DeepSeek-R1 (49.85%) and QwQ-32B (47.09%). Weaker models like DeepSeek-MOE-16B-Chat (4.71%) and Mistral-Nemo-Instruct-2407 (8.01%) struggle across all levels, particularly at UG/PG Phys, underscoring their inadequacy for modern physics. The High School Olympiad (HSO) and UG Non-Physics levels show moderate performance, with top models like o3 (high) (71.81% HSO,

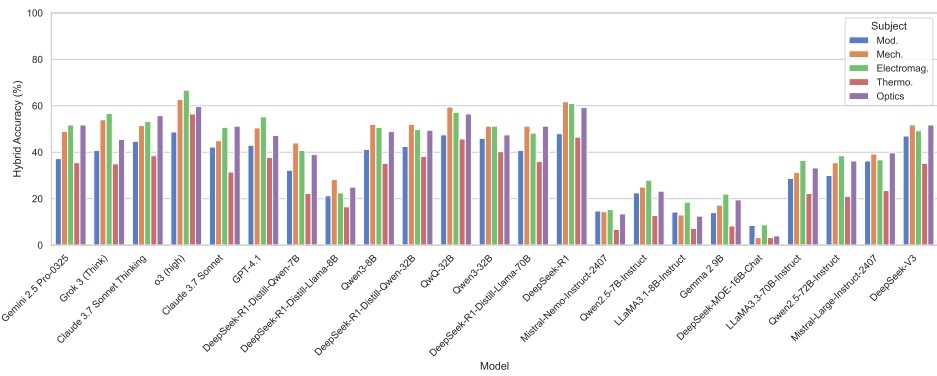

Figure 10: Model Performance by Subject

73.10% UG Non-Phys) maintaining relatively high accuracies, while others, such as DeepSeek-R1-Distill-Llama-8B (29.59% HSO, 34.21% UG Non-Phys), lag. This gradient in performance highlights the increasing complexity of physics problems and the superior reasoning capabilities of top models.

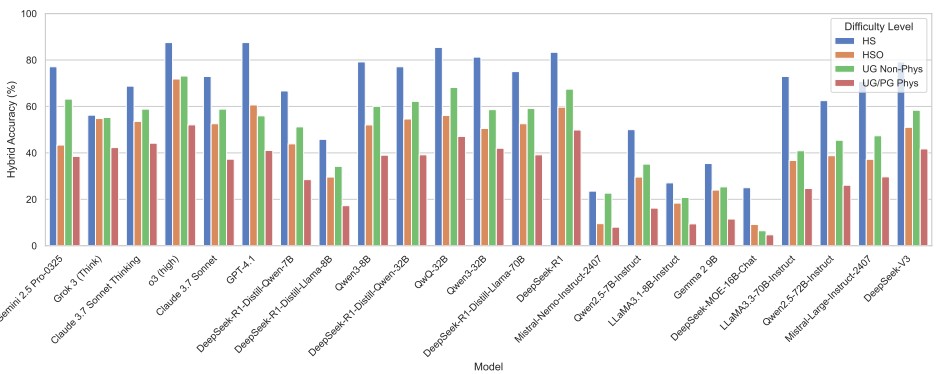

Figure 11: Model Performance by Difficulty Level

## F.4  Comparison with Existing Datasets

We briefly explain how our PHYSICS dataset differs from existing physics-related datasets. Our contributions to data are primarily reflected in two areas: physics training data and physics test data.

### F.4.1  Physics Training Data

**Motivation.** Current LLMs exhibit insufficient physics capabilities, yet there is a lack of high-quality training datasets for physics, making it difficult to effectively improve models in this domain.

**Contribution.** We created the first high-quality physics training dataset at a scale of 14,568 samples.

• The dataset spans a wide range of difficulty levels: High School and below, High School Olympiad, Non-Physics Undergraduate, and Undergraduate/Postgraduate (Physics Major), encompassing most stages of physics education and providing a comprehensive coverage of various academic levels.

• It includes comprehensive coverage across five major areas of physics: Modern Physics, Mechanics, Electromagnetism, Thermodynamics, and Optics.

• A strict quality control process, combining LLM-based evaluations, expert reviews, and systematic cross-checking, ensures data reliability and consistency throughout the entire dataset creation process.

• For user convenience, we provide 4,000 samples with detailed reasoning paths generated by QwQ-32B, and we plan to release more reasoning paths using stronger models (e.g., DeepSeek-R1, Qwen3-235B-A22B, etc.) in the future.

Table 16: The subject distribution of PHYSICS test set. Subjects are divided Mod.(Modern Physics), Mech. (Mechanics), Electromag.(Electromagnetism), Thermo. (Thermodynamics), and Optics.

| Benchmark | Mod. | Mech. | Electromag. | Thermo. | Optics. | Total |
|---|---|---|---|---|---|---|
| MMLU | 114 | 177 | 94 | 52 | 51 | 488 |
| GPQA | 191 | 10 | 15 | 4 | 7 | 227 |
| Olympiadbench | 79 | 136 | 52 | 71 | 13 | 351 |
| UGPhysics | 4998 | 3430 | 1148 | 744 | 720 | 11040 |
| PHYSICS(ours) | 400 | 400 | 400 | 400 | 400 | 2000 |

Table 17: The difficulty distribution of our PHYSICS test set. Difficulty levels include: 1: High School and Below, 2: High School Olympiad, 3: Undergraduate (Non-Physics Major), 4: Undergraduate/Postgraduate(Physics Major).

| Benchmark | 1 | 2 | 3 | 4 | Total |
|---|---|---|---|---|---|
| MMLU | 386 | 0 | 102 | 0 | 488 |
| GPQA | 0 | 0 | 41 | 186 | 227 |
| Olympiadbench | 0 | 351 | 0 | 0 | 351 |
| UGPhysics | 0 | 0 | 1463 | 9577 | 11040 |
| PHYSICS(ours) | 48 | 196 | 418 | 1338 | 2000 |

### F.4.2 Physics Test Data

**Motivation.** As shown in Tab. 1, existing physics test sets have imbalanced coverage in both difficulty and subject areas, limiting their ability to comprehensively evaluate a model's physics capabilities.

**Contribution.**

• We constructed the first dataset that comprehensively spans the full range of physics problems, including high school physics, Olympiad-level physics, Undergraduate (Non-Physics Major) physics, and Undergraduate/Postgraduate(Physics Major) physics.

• Moreover, our dataset is the first in the field to ensure a uniform distribution across physics subjects. As shown in the experimental results in Table 7 of the paper, model performance varies significantly across different subjects. A balanced subject distribution therefore makes the average benchmark score more representative and fair.

• We performed strict quality control on the test data to ensure high data quality.

**Data Analysis.** In Tab. 16 and Tab. 17, we present a detailed comparison of the difficulty and subject distribution of physics-related plain-text test sets across PHYSICS and MMLU, GPQA, OlympiadBench, and UGPhysics, highlighting key differences.

Here, we also present a comparison of evaluation results on plain-text physics questions from UGPhysics, OlympiadBench, and GPQA. To avoid excessive testing consumption and to better evaluate the model's physical reasoning capabilities, UGPhysics samples 2,000 instances by topic, language, and level, preserving the original distribution (all other references to UGPhysics in the rebuttal and the main paper refer to the full dataset). Only physics questions from GPQA and OlympiadBench are used. This version includes models that performed well on PHYSICS and UGPhysics; a full comparison will appear in the next version. Results are in Tab. 18.

Our two main contributions compared to existing test sets are a **balanced subject and difficulty distribution** and **high-quality data**.

Balanced subject and difficulty distribution. From the results shown in the tables, we can observe that our test set ensures a balanced distribution not only in terms of physics subfields but also in difficulty levels, which span high school, high school competitions, undergraduate, and physics major (graduate-level) problems. Both aspects are crucial for comprehensive evaluation.

As shown in Tab. 7, model performance varies significantly across different subfields. A balanced subject distribution ensures average scores better reflect overall physics ability. Similarly, a balanced

Table 18: Cross-benchmark comparison.

| Model | PHYSICS | | UGPhysics | | OlympiadBench | | GPQA | |
|---|---|---|---|---|---|---|---|---|
| | Rule | Hybrid | Rule | Hybrid | Rule | Hybrid | Rule | Hybrid |
| o3-2025-04-16 | 23.30 | 58.90 | 28.20 | 45.05 | 53.28 | 66.95 | 25.11 | 85.02 |
| DeepSeek-R1 | 27.55 | 55.30 | 32.05 | 40.75 | 50.43 | 61.54 | 37.44 | 79.30 |
| Claude-3-7-sonnet-thinking | 21.60 | 48.75 | 28.75 | 38.25 | 49.29 | 58.40 | 44.49 | 81.94 |
| QwQ-32B | 20.65 | 53.30 | 28.55 | 39.35 | 50.43 | 63.53 | 47.58 | 75.37 |
| DeepSeek-V3 | 22.45 | 47.05 | 27.80 | 36.05 | 46.15 | 54.42 | 44.05 | 66.52 |
| gpt-4.1 | 21.30 | 46.75 | 25.35 | 36.40 | 48.72 | 56.98 | 23.79 | 75.33 |

difficulty distribution assesses both reasoning and knowledge. As discussed in Sec. 4.1, High school and Olympiad problems test reasoning, while undergraduate and graduate problems focus on content knowledge. Covering all difficulty levels enables a thorough evaluation.

High-quality data. Among existing datasets, UGPhysics is most similar to ours but with significantly lower quality. From 2,000 UGPhysics samples, 789 questions were unanswered by six models (Table 2.3). Manual review found only 19.39% valid, 52.75% flawed questions, and 27.88% flawed answers. These issues are well addressed in our dataset.

Common question issues include: (1) Reliance on previous context (i.e., those commonly omitted in standard physics problems), (2) Missing key conditions (i.e., those commonly omitted in standard physics problems), (3) Extraction errors (e.g., truncation, missing symbols, incoherent text), (4) References to unavailable figures.

Answer issues include: (1) Incomplete responses, (2) Incorrect final answers or explanations, (3) Invalid reasoning in open-ended questions.

In short, our dataset's higher quality ensures more trustworthy model evaluation results. A more detailed analysis will be presented in the next version of the paper.

In summary, compared to existing test sets, ours better captures the multifaceted nature of a model's physics proficiency. It is not merely an incremental addition but a well-constructed and comprehensive benchmark dataset.

# G Case Study

## G.1 Model Inference

We demonstrate cases of model inference, including the question, solution, and answer from the original data, as well as the test result derived from model inference. The cases are presented in two types: Chinese and English questions.

---

### English Question Inference

**id:** 2852

**question:** Which type of particle emits the strongest synchrotron radiation when bending in a magnetic field? (A) Proton (B) $\mu$ Meson (C) Electron (D) $\pi$ Meson.

**solution:** Solution: From the result of problem 17.10, the energy of synchrotron radiation emitted by a particle during one revolution in a magnetic field is

$$\delta E = \frac{4\pi}{3}\frac{e^2}{\rho}\beta^3\gamma^4 = \frac{4\pi}{3}\frac{e^2}{\rho}\beta^3\left(\frac{E}{m}\right)^4$$

where $m$ and $e$ are the mass and charge of the particle, respectively, and $\rho$ is the radius of the trajectory. The smaller the particle's mass, the stronger the synchrotron radiation. Obviously, the answer is (C).

**answer:** (C)

**answer_type:** ["MCQ"]

**language:** en

**domain:** Advanced Physics

**difficulty:** Physics UnderGraduate

**translate:** true

**test_result:** To determine which type of particle emits the strongest synchrotron radiation when bending in a magnetic field, we need to understand that the intensity of synchrotron radiation is proportional to the square of the charge and inversely proportional to the mass of the particle.

The formula for the power radiated per unit solid angle (intensity) due to synchrotron radiation is given by:

$$P = \frac{\epsilon_0 c e^4}{6\pi m^2}\beta^4\gamma^4\left(\frac{d\vec{\beta}}{dt}\right)^2$$

where: - $e$ is the elementary charge, - $m$ is the rest mass of the particle, - $\beta = v/c$ (the velocity divided by the speed of light), - $\gamma = 1/\sqrt{1-\beta^2}$ (the Lorentz factor).

For particles with similar velocities ($\beta$ and $\gamma$ values), the key factors affecting the intensity of synchrotron radiation are the charge squared over the mass ($e^2/m$). Since all these particles have the same charge magnitude $|e|$, we only need to compare their masses.

Here are the approximate masses of the particles involved: - Proton: $m_p \approx 938$ MeV$/c^2$ - Muon ($\mu$ meson): $m_\mu \approx 105.7$ MeV$/c^2$ - Electron: $m_e \approx 0.511$ MeV$/c^2$ - Pi meson ($\pi$ meson): $m_\pi \approx 139.6$ MeV$/c^2$

Given these masses, the electron has the smallest mass among the listed particles. Therefore, it will emit the most intense synchrotron radiation because the ratio $e^2/m$ is largest for the electron. Thus, the correct answer is:

\boxed{C}

---

## G.2 Evaluation Optimization

Cases we mentioned in Section 3.3 now can be judged correctly by Physics-xVerify.

*Case 1: Unit Conversion, but the unit is not explicitly specified*

**Question:** In an experiment, the wavelength of light is measured using single-slit diffraction. The slit width is $a = 0.2\,\text{mm}$, the distance from the slit to the screen is $L = 1.5\,\text{m}$, and the distance from the first dark fringe to the central bright fringe on the screen is $x = 4.5\,\text{mm}$. Find the wavelength of light $\lambda$.
**Ground Truth:** $0.6 \times 10^{-6} m$
**Model Output:** $600 nm$
**Previous Judgement:** Incorrect
**Physics-xVerify Judgement:** Correct

# H  Limitations and Future Work

The dataset constructed in this paper currently focuses only on text-based questions, while multi-modal problems are also common in physics-related data. Therefore, we plan to release a multi-modal version of the dataset as the next iteration of PHYSICS. In addition, we will continue to provide reasoning paths of various models on the training set, aiming to further assist models in learning physics knowledge and improving their physical reasoning abilities, thereby pushing the limits of model intelligence of Artificial Intelligence.

# I  Broader Impacts

We construct the largest-scale, most comprehensive, and high-quality physics dataset, PHYSICS, covering multiple sub-disciplines of physics. This dataset can be used not only for training models to improve their physics-related capabilities but also for evaluating the physical reasoning abilities of current models. In addition, we design a specialized evaluation framework tailored to physics-based problems. We hope that through the dataset and evaluation methodology we provide, we can promote the development of large language models in the field of physics, enabling them to understand and apply physics principles, thereby pushing forward the upper bound of model capabilities and better assisting humans in real-world applications.

