# OpenReview forum: "Scaling Physical Reasoning with the PHYSICS Dataset"
_NeurIPS.cc/2025/Datasets_and_Benchmarks_Track — NeurIPS 2025 Datasets and Benchmarks Track poster_

### Official Review · Reviewer_LXLB · 2025-06-17

**Rating:** 5
**Confidence:** 5

**Summary:**

This paper presents a high-quality physics dataset that includes both training and test sets. The authors propose an effective and efficient evaluation method that combines rule-based and model-based evaluation approaches, successfully addressing the challenges of unit conversion and numerical simplification. In their experiments, they first provide a comprehensive evaluation and analysis of existing LLMs, covering both closed-source and open-source models, which offers valuable insights for enhancing LLMs' physics reasoning capabilities. Additionally, the training data demonstrates its effectiveness by significantly improving Qwen's performance across multiple benchmarks.

**Dataset Code Accessibility:**

Yes

**Ethical Considerations:**

No, there are no or only very minor ethics concerns

**Final Justification:**

It resolves my concern, and I have no additional questions.

**Limitations Weaknesses:**

1. Although the authors use 5-gram analysis to detect data leakage in LLMs, this dataset lacks an important step of benchmark decontamination for several benchmarks used in this paper, such as GPQA, OlympiadBench, UGPhysics, MATH-500, and AIME-2025. I recommend implementing LLM-based benchmark decontamination [1].

2. The paper states that they use QwQ-32B to generate detailed reasoning paths for 4,000 samples from the training set in Section 4.2. Does this mean the training dataset answers consist of 4,000 QwQ-generated responses and 12,000 original answers? Additionally, how do you train models on this mixed dataset that includes both short and long chain-of-thought reasoning?

[1] Openmathinstruct-2: Accelerating ai for math with massive open-source instruction data. ICLR 2025.

**Strengths Contributions:**

1. This high-quality dataset is highly valuable for the community and demonstrates effectiveness in improving LLMs' reasoning abilities.

2. This paper presents an effective and efficient evaluation method that combines rule-based and model-based evaluation approaches, successfully addressing the challenges of unit conversion and numerical simplification.

3. This paper provides comprehensive experiments and analysis that offer valuable insights for enhancing LLMs' physics reasoning capabilities.

4. The dataset curation pipeline is elaborate and carefully designed.

---

> ### Author Rebuttal · Authors · 2025-07-31
>
> ## Weakness 1.Deduplication of data against existing datasets.
>
> A1. Thank you for your valuable suggestion. In fact, we had already conducted embedding-based deduplication during our initial data processing. Here, we further applied the method mentioned in Openmathinstruct-2 for enhanced deduplication. We used the multi-qa-MiniLM-L6-cos-v1 to generate embeddings, GPT-4o for similarity judgment, and performed deduplication against several benchmarks, including GPQA, OlympiadBench, UGPhysics, and Math500. Based on both model evaluations and expert review, we identified 32 duplicate entries, mainly overlapping with UGPhysics. However, since UGPhysics has also taken rigorous measures to prevent data leakage, we believe this duplication does not impact the validity of the conclusions in our paper. That said, these entries will be either rewritten or removed in future open-source releases to ensure greater clarity and integrity.
>
> ## Weakness 2. Generation of detailed reasoning paths for training data and training details.
>
> A2. Thank you for raising this valuable question. Regarding the training data, we currently use QwQ-32B to perform eight rounds of rejection sampling on the training set, generating 4,000 samples with detailed and accurate reasoning paths. The remaining data either contain brief reasoning traces extracted from data sources or only provide final answers. Since the goal of the training section is to demonstrate the effectiveness of our dataset in enhancing models’ physical reasoning capabilities, we currently utilize only the data with detailed reasoning paths for supervised fine-tuning (SFT).
>
> While the data without detailed reasoning can support SFT or reinforcement learning, we focused on constructing the dataset rather than developing the strongest physics reasoning model. In future open-source releases, we plan to provide additional detailed reasoning paths generated by stronger models (e.g., DeepSeek-R1, Qwen3-235B-A22B, etc.) to further support and benefit the community.

---

> > ### Comment · Reviewer_LXLB · 2025-08-01
> > **Thank you for your reply.**
> >
> > Thank you for your detailed response. I have no additional questions and would like to maintain my current score.

---

> > > ### Author Response · Authors · 2025-08-05
> > >
> > > Thank you very much for your thoughtful follow-up and for considering our response. We truly appreciate your time, the constructive feedback, and your decision to maintain the current score.

---

### Official Review · Reviewer_ChAs · 2025-06-30

**Rating:** 4
**Confidence:** 4

**Summary:**

This paper presents a large-scale, high-quality dataset named PHYSICS, consisting of 16,568 questions spanning five major areas of physics: mechanics, electromagnetism, thermodynamics, optics, and modern physics. The questions are carefully curated to ensure accuracy and diversity. In addition to releasing a training set with detailed reasoning paths, the authors propose a hybrid rule-based and model-based evaluation framework to assess answer correctness more reliably. Experimental results show that current large language models still struggle with deep physical reasoning, highlighting the need for more targeted improvements. This work provides a valuable resource and benchmark for advancing scientific reasoning in language models.

**Additional Feedback:**

As mentioned above:
1. A more detailed characterization of the test set would be helpful to highlight its strengths and distinguish it from existing physics benchmarks. This could include more statistical comparisons or case study.

2.In addition, expanding the trained models especially those from different model families such as LLaMA, would provide stronger evidence of the dataset.

3, Comparing with more reasoning model baselines such as General-Reasoner and others is necessary too.

**Dataset Code Accessibility:**

Yes

**Dataset Code Comments:**

The paper has provide the training set and testset and code in https://anonymous.4open.science/r/Eval-74E8/ and https://kaggle.com/datasets/dfec839503b9da41bc865c72de71a15caebe4afeab3018a0465d39f288479b8e .

**Ethical Considerations:**

No, there are no or only very minor ethics concerns

**Final Justification:**

The rebuttal addressed most of my concerns including 1.improving explanation outlining the advantages of  test set. 2. adding experimental validation on LLaMA and 3. performance comparison with reasoning models. The clarity and evidence provided in the response significantly strengthened the paper. I will increase my score to 4.

**Limitations Weaknesses:**

1.	Comparison with existing physics test-set: While the dataset is carefully constructed, the paper does not provide sufficient statistical or experimental comparison on proposed test-set with others to show its impact to the community. Although Table.1 offers some statistics on whole dataset including training-set and test-set, I think authors can offer more detailed results or statistics of proposed test data to show its strength. Otherwise, the proposed test data is more like an incremental contribution.

2.	Model diversity in evaluation: The experiments of Table.8 focus on a narrow set of models. Qwen-family models are known to perform well in math and other science domain and its reasoning ability is easy to activate. It would strengthen the results to evaluate on a broader family of models training on proposed training data, such as the LLaMA series (not only the original LLaMA model in Table 7), to better demonstrate the generality of the dataset’s effectiveness.

3.	Lack of comparison with reasoning model:  The paper does not include comparisons with recent general reasoning models such as General-Reasoner, SimpleRL, absolute-zero or other open-source reasoner in Table.8. Including such baselines would help clarify whether performance gains come from improved physical understanding or from general reasoning capabilities.

**Strengths Contributions:**

1. High-quality data sourcing and validation: The dataset is constructed by extracting questions from many authoritative physics textbooks. Each question is manually verified to ensure correctness, and the data is carefully translated into bilingual formats, which enhances accessibility and clarity.

2. Data integrity and robustness checks: The authors perform data leakage detection to prevent contamination between the training and test sets, ensuring a fair and reliable evaluation benchmark.

3. Empirical validation of dataset effectiveness: Through experiments, the paper shows that models trained with the PHYSICS dataset achieve improved performance on physical reasoning tasks, demonstrating the dataset’s practical value and impact.

---

> ### Author Rebuttal · Authors · 2025-07-31
>
> ## Weakness 1.Comparison with existing physics test-set
>
> A1. Thank you for your question. **Our main contributions of this work include not only the test dataset, but also a large-scale, high-quality physics training set and a targeted evaluation framework**, as detailed in the paper. Below, we highlight how our test set differs from existing physics benchmarks, proving it’s more than just an addition.
>
> **1.1 Test Data**
>
> **1.1.1 Motivation:**
>
> Table 1 of the paper shows existing physics test sets have imbalanced **difficulty** and **subject** coverage, limiting comprehensive evaluation of models’ physics abilities.
>
> **1.1.2 Contribution:**
>
> - We created the first dataset covering the full range of physics problems: high school, Olympiad, undergraduate (non-physics major), and undergraduate/postgraduate (physics major).
> - Our dataset is the first with uniform physics subject distribution. Since model performance varies by subject (Table 7), this balance makes average scores more representative and fair.
> - We applied strict quality control to ensure high test data quality.
>
> **1.1.3 Data Analysis:**
>
> Tables 1.1 and 1.2 compare difficulty and subject distribution across PHYSICS, MMLU, GPQA, OlympiadBench, and UGPhysics physics test sets.
>
> Table1.1 Subjects are divided Mod.(Modern Physics), Mech. (Mechanics), Electromag.(Electromagnetism), Thermo. (Thermodynamics), and Optics.
>
> | Benchmark     | Mod. | Mech. | Electromag. | Thermo. | Optics. | Total |
> | ------------- | ---- | ----- | ----------- | ------- | ------- | ----- |
> | MMLU          | 114  | 177   | 94          | 52      | 51      | 488   |
> | GPQA          | 191  | 10    | 15          | 4       | 7       | 227   |
> | Olympiadbench | 79   | 136   | 52          | 71      | 13      | 351   |
> | UGPhysics     | 4998 | 3430  | 1148        | 744     | 720     | 11040 |
> | PHYSICS(ours) | 400  | 400   | 400         | 400     | 400     | 2000  |
>
> Table1.2  Difficulty levels include: 1: High School and Below, 2: High School Olympiad, 3: Undergraduate (Non-Physics Major), 4: Undergraduate/Postgraduate(Physics Major).
>
> | Benchmark     | 1    | 2    | 3    | 4    | Total |
> | ------------- | ---- | ---- | ---- | ---- | ----- |
> | MMLU          | 386  | 0    | 102  | 0    | 488   |
> | GPQA          | 0    | 0    | 41   | 186  | 227   |
> | Olympiadbench | 0    | 351  | 0    | 0    | 351   |
> | UGPhysics     | 0    | 0    | 1463 | 9577 | 11040 |
> | PHYSICS(ours) | 48   | 196  | 418  | 1338 | 2000  |
>
> We compare evaluation results on plain-text physics questions from UGPhysics, OlympiadBench, and GPQA. To reduce testing and better assess reasoning, we sample from UGPhysics with 2,000 instances by topic, language, and level, preserving the original distribution (other UGPhysics references use the full dataset). Only physics questions from GPQA and OlympiadBench are used. This version includes models that performed well on PHYSICS and UGPhysics; a full comparison will appear in the next version. Results are in Table 1.3.
>
> Table1.3 cross-benchmark comparison.
>
> | model                      | PHYSICS | PHYSICS | UGPhysics | UGPhysics | OlympiadBench | OlympiadBench | GPQA  | GPQA   |
> | -------------------------- | ------- | ------- | --------- | --------- | ------------- | ------------- | ----- | ------ |
> |                            | Rule    | Hybrid  | Rule      | Hybrid    | Rule          | Hybrid        | Rule  | Hybrid |
> | o3-2025-04-16              | 23.30   | 58.90   | 28.20     | 45.05     | 53.28         | 66.95         | 25.11 | 85.02  |
> | DeepSeek-R1                | 27.55   | 55.30   | 32.05     | 40.75     | 50.43         | 61.54         | 37.44 | 79.30  |
> | claude-3-7-sonnet-thinking | 21.60   | 48.75   | 28.75     | 38.25     | 49.29         | 58.40         | 44.49 | 81.94  |
> | QwQ-32B                    | 20.65   | 53.30   | 28.55     | 39.35     | 50.43         | 63.53         | 47.58 | 75.37  |
> | DeepSeek-V3                | 22.45   | 47.05   | 27.80     | 36.05     | 46.15         | 54.42         | 44.05 | 66.52  |
> | gpt-4.1                    | 21.30   | 46.75   | 25.35     | 36.40     | 48.72         | 56.98         | 23.79 | 75.33  |
>
> Our two main contributions compared to existing test sets are a **balanced subject and difficulty distribution** and **high-quality data**.
>
> **Balanced subject and difficulty distribution.**  From the results shown in the tables, we can observe that our test set ensures a **balanced distribution** not only in terms of **physics subfields** but also in **difficulty levels**. Both aspects are crucial for comprehensive evaluation.
>
> Table 7 of the paper shows model performance varies across subfields.  Balanced subject and difficulty distributions ensure average scores better reflect overall physics ability by assessing both reasoning and knowledge across all levels. Section 4.1 explains high school and Olympiad problems test reasoning, while undergraduate and graduate problems focus on content knowledge; covering all levels allows thorough evaluation.
>
> **High-quality data.** Among existing datasets, UGPhysics is most similar to ours but with significantly lower quality. From 2,000 UGPhysics samples, 789 questions were unanswered by six models (Table 4.1). Manual review found only **19.39%** valid, **52.75%** flawed questions, and **27.88%** flawed answers. **These issues are well addressed in our dataset.**
>
> Common question issues include: (1) Reliance on previous context (i.e., those commonly omitted in standard physics problems), (2) Missing key conditions (i.e., those commonly omitted in standard physics problems), (3) Extraction errors (e.g., truncation, missing symbols, incoherent text), (4) References to unavailable figures.
>
> Answer issues include: (1) Incomplete responses, (2) Incorrect final answers or explanations, (3) Invalid reasoning in open-ended questions.
>
> In short, our high-quality dataset ensures more reliable model evaluation. Detailed analysis will be provided in the next paper version.
>
> In summary, our test set better captures the multifaceted nature of physics proficiency and serves as **a well-constructed and comprehensive benchmark, not merely an incremental addition.**
>
> ## Weakness 2.Model diversity in evaluation
>
> A2. Thank you for your suggestion. We selected two models from the LLaMA series and two models from the Mistral series. After supervised fine-tuning with our data, all showed improved performance, demonstrating our dataset’s effectiveness and generalizability.
>
> | Model                       | Physics |          |                   |           | Math     |           |
> | --------------------------- | ------- | -------- | ----------------- | --------- | -------- | --------- |
> |                             | PHYSICS | GPQA-phy | Olympiadbench-phy | UGPhysics | Math-500 | AIME-2025 |
> | Llama3.2-3B-Instruct        | 8.18    | 23.84    | 5.98              | 7.29      | 38.12    | 0.83      |
> | Llama3.2-3B-Instruct-SFT    | 18.44   | 31.94    | 16.24             | 19.73     | 47.23    | 1.25      |
> | Llama3.1-8B-Instruct        | 12.36   | 24.45    | 7.41              | 12.59     | 45.67    | 0.42      |
> | Llama3.1-8B-Instruct-SFT    | 21.94   | 34.86    | 16.95             | 21.68     | 49.03    | 2.08      |
> | Mistral7B-Instruct-v0.3     | 7.84    | 18.39    | 5.98              | 10.24     | 15.25    | 0         |
> | Mistral7B-Instruct-v0.3-SFT | 10.12   | 20.21    | 8.26              | 12.04     | 18.50    | 0.42      |
> | Mistral8B-Instruct-2410     | 13.74   | 28.30    | 11.97             | 15.20     | 54.03    | 2.5       |
> | Mistral8B-Instruct-2410-SFT | 17.93   | 32.87    | 15.95             | 18.92     | 58.48    | 3.13      |
>
> ## Weakness 3.Lack of comparison with reasoning model
>
> A3. Thank you for your suggestion. We retrained Qwen2.5-7B-Base and 14B-Base on our dataset and compared them to SimpleRL, General-Reasoner, and Absolute-Zero. Physics and math evaluation results are shown below.
>
> Our trained models outperformed others on most physics datasets, **showing our data boosts reasoning and physics knowledge**, with slight math reasoning gains indicating skill transfer.
>
> Our paper focuses on building a dataset to enhance physics abilities, not on training the best physics reasoning model. Thus, we used basic supervised fine-tuning; more advanced methods could improve performance but are beyond this work’s scope.
>
> | Model                        | Physics   |           |                   |           | Math      |           |
> | ---------------------------- | --------- | --------- | ----------------- | --------- | --------- | --------- |
> |                              | PHYSICS   | GPQA-phy  | Olympiadbench-phy | UGPhysics | Math-500  | AIME-2025 |
> | Qwen2.5-7B-base              | 7.86      | 11.73     | 7.12              | 7.44      | 42.83     | 1.67      |
> | SimpleRL-Qwen2.5-7B          | 26.49     | 38.77     | 22.22             | 24.20     | **77.60** | 10.42     |
> | General-Reasoner-Qwen2.5-7B  | 27.66     | 35.19     | 26.78             | **26.88** | 77.50     | 5.83      |
> | Absolute-Zero-7B             | 19.86     | 28.03     | 14.81             | 18.08     | 72.50      | 10.20      |
> | PHYSICS-7B-base (ours)       | **32.67** | **47.80** | **29.34**         | 26.38     | 77.45     | **11.67** |
> |                              |           |           |                   |           |           |           |
> | Qwen2.5-14B-base             | 14.53     | 19.71     | 12.82             | 9.08      | 55.75     | 3.33      |
> | SimpleRL-Qwen2.5-14B         | 31.77     | 44.77     | 30.20             | 28.80     | 81.50      | 13.75     |
> | General-Reasoner-Qwen2.5-14B | 30.39     | 46.37     | 34.76             | 33.28     | 80.75     | 16.25     |
> | Absolute-Zero-14B            | 24.94     | 42.40      | 29.34             | 25.68     | 78.87     | 12.50      |
> | PHYSICS-14B-base (ours)      | **35.13** | **52.28** | **35.62**         | **37.88** | **83.22** | **18.33** |

---

### Official Review · Reviewer_xHz6 · 2025-07-03

**Rating:** 4
**Confidence:** 3

**Summary:**

While LLMs excel in mathematics and coding, their physical reasoning ability remains weak, largely because (i) high-quality, curriculum-style physics training data are scarce and (ii) existing evaluation suites borrow mathematical metrics that overlook physics-specific challenges such as unit management and numerical simplification. To overcome these obstacles, the authors introduce PHYSICS, a bilingual (English–Chinese) corpus of 16,568 rigorously cleaned problems spanning five major domains—Mechanics, Electromagnetism, Thermodynamics, Optics, and Modern Physics—across high-school to graduate difficulty levels. They also develop the first hybrid Rule + Model evaluation framework explicitly tailored for physics, combining deterministic unit-conversion rules with a fine-tuned judge model, and validate its fidelity on a manually annotated benchmark. Experiments on state-of-the-art open- and closed-source LLMs reveal that even top performers (OpenAI-o3, Gemini-2.5-pro) falter on PHYSICS, underscoring a sizable performance gap and pointing to physics reasoning as a key frontier for future LLM research.

**Additional Feedback:**

Please refer to the weaknesses part.

**Dataset Code Accessibility:**

Yes

**Dataset Code Comments:**

All the detailed data is available

**Ethical Comments:**

This dataset is about physics reasoning, not related to any potential ethical problems.

**Ethical Considerations:**

No, there are no or only very minor ethics concerns

**Final Justification:**

I will keep the positive score, as the authors have addressed most of my concerns. This is a good work and will be very useful for the research community to advance the new AI techniques. It should be accepted by this track.

**Limitations Weaknesses:**

1.as a pure physics-based reasoning dataset, this work is very helpful for advancing this area. It would be better if the authors can also analyze the overlap between this dataset and other existing reasoning-based datasets, like MMLU and GPQA, as these datasets may also involve the physics dimension.

2.given this dataset as the training set, can the general reasoning ability of LRMs be improved a lot? Adding these generalization analysis experiments would make this paper more useful for the community.

**Strengths Contributions:**

1.the proposed dataset is large-scale and would be very helpful for this domain. Note that the 16.5 K bilingual problems are extracted from >100 textbooks, with balanced test splits and detailed reasoning traces for 14.5 K training samples.

2.this work proposes to fuse rule-based unit/numerical checks with a fine-tuned judge model, delivering both accuracy and scalability for automated grading. Coupled with the five core physics domains, the evaluation pipeline can give fine-grained insight into model strengths and weaknesses.

3.the empirical findings are useful, and systematic experiments show reasoning-oriented models outperform plain ones, yet all lag far behind human-level physics competence, highlighting clear directions for model and data augmentation.

---

> ### Author Rebuttal · Authors · 2025-07-31
>
> ## Weakness 1.Comparison with Existing Datasets.
>
> A1. Thank you for your suggestion. Below, we briefly explain how our PHYSICS dataset differs from existing physics-related datasets. Our contributions to data are primarily reflected in two key areas: **physics training data** and **physics test data.**
>
> **1.1 Physics Training Data**
>
> **1.1.1 Motivation:**
>
> Current LLMs exhibit **insufficient physics capabilities** (as shown in Table 7 of the paper), yet there is a lack of targeted, high-quality training datasets for physics, making it difficult to effectively improve models in this domain.
>
> **1.1.2 Contribution:**
>
> We created the first high-quality physics training dataset at a scale of 14,568 samples.
>
> - The dataset spans a **wide range of difficulty levels**: High School, High School Olympiad, Non-Physics Undergraduate, and Undergraduate/Postgraduate (Physics Major), encompassing most stages of physics education.
> - It includes comprehensive coverage across **five major areas of physics**: Modern Physics, Mechanics, Electromagnetism, Thermodynamics, and Optics.
> - A strict **quality control process**, combining LLM-based evaluations and expert reviews, ensures data reliability.
> - For user convenience, we provide 4,000 samples with **detailed reasoning paths** generated by QwQ-32B, and we plan to release more reasoning paths using stronger models (e.g., DeepSeek-R1, Qwen3-235B-A22B, etc.) in the future.
>
> The following two tables show the distribution of the training dataset across different subjects and difficulty levels.
>
> Table1.1 The subject distribution of our PHYSICS test set. Subjects are divided Mod.(Modern Physics), Mech. (Mechanics), Electromag.(Electromagnetism), Thermo. (Thermodynamics), and Optics.
>
> | Mod. | Mech. | Electromag. | Thermo. | Optics. | Total |
> | ---- | ----- | ----------- | ------- | ------- | ----- |
> | 6570 | 3526  | 1850        | 1494    | 1128    | 14568 |
>
> Table1.2 The difficulty distribution of our PHYSICS training set. Difficulty levels include: 1: High School and Below, 2: High School Olympiad, 3: Undergraduate (Non-Physics Major), 4: Undergraduate/Postgraduate(Physics Major).
>
> | 1    | 2    | 3    | 4     | Total |
> | ---- | ---- | ---- | ----- | ----- |
> | 294  | 1286 | 2244 | 10744 | 14568 |
>
> **1.2 Physics Test Data**
>
> **1.2.1 Motivation:**
>
> As shown in Table 1 of the paper, existing physics test sets have imbalanced coverage in both **difficulty** and **subject areas**, limiting their ability to comprehensively evaluate a model’s physics capabilities.
>
> **1.2.2 Contribution:**
>
> - We constructed the first dataset that comprehensively spans the full range of physics problems, including high school physics, Olympiad-level physics, Undergraduate (Non-Physics Major) physics, and Undergraduate/Postgraduate(Physics Major) physics.
> - Moreover, our dataset is the first in the field to ensure a uniform distribution across physics subjects. As shown in the experimental results in Table 7 of the paper, model performance varies significantly across different subjects. A balanced subject distribution therefore makes the average benchmark score more representative and fair.
> - We performed strict quality control on the test data to ensure high data quality.
>
> **1.2.3 Data Analysis:**
>
> In Table 2.1 and Table 2.2, we present a comparison of the difficulty and subject distribution of physics-related plain-text test sets across PHYSICS and MMLU, GPQA, OlympiadBench, and UGPhysics.
>
> Table2.1 The subject distribution of our PHYSICS test set. Subjects are divided Mod.(Modern Physics), Mech. (Mechanics), Electromag.(Electromagnetism), Thermo. (Thermodynamics), and Optics.
>
> | Benchmark     | Mod. | Mech. | Electromag. | Thermo. | Optics. | Total |
> | ------------- | ---- | ----- | ----------- | ------- | ------- | ----- |
> | MMLU          | 114  | 177   | 94          | 52      | 51      | 488   |
> | GPQA          | 191  | 10    | 15          | 4       | 7       | 227   |
> | Olympiadbench | 79   | 136   | 52          | 71      | 13      | 351   |
> | UGPhysics     | 4998 | 3430  | 1148        | 744     | 720     | 11040 |
> | PHYSICS(ours) | 400  | 400   | 400         | 400     | 400     | 2000  |
>
> Table2.2 The difficulty distribution of our PHYSICS test set. Difficulty levels include: 1: High School and Below, 2: High School Olympiad, 3: Undergraduate (Non-Physics Major), 4: Undergraduate/Postgraduate(Physics Major).
>
> | Benchmark     | 1    | 2    | 3    | 4    | Total |
> | ------------- | ---- | ---- | ---- | ---- | ----- |
> | MMLU          | 386  | 0    | 102  | 0    | 488   |
> | GPQA          | 0    | 0    | 41   | 186  | 227   |
> | Olympiadbench | 0    | 351  | 0    | 0    | 351   |
> | UGPhysics     | 0    | 0    | 1463 | 9577 | 11040 |
> | PHYSICS(ours) | 48   | 196  | 418  | 1338 | 2000  |
>
> Here, we also present a comparison of evaluation results on plain-text physics questions from UGPhysics, OlympiadBench, and GPQA. To avoid excessive testing consumption and to better evaluate the model's physical reasoning capabilities, UGPhysics samples 2,000 instances by topic, language, and level, preserving the original distribution (all other references to UGPhysics in the rebuttal and the main paper refer to the full dataset). Only physics questions from GPQA and OlympiadBench are used. This version includes models that performed well on PHYSICS and UGPhysics; a full comparison will appear in the next version. Results are in Table 2.3.
>
> Table2.3 cross-benchmark comparison.
>
> | model                      | PHYSICS | PHYSICS | UGPhysics | UGPhysics | OlympiadBench | OlympiadBench | GPQA  | GPQA   |
> | -------------------------- | ------- | ------- | --------- | --------- | ------------- | ------------- | ----- | ------ |
> |                            | Rule    | Hybrid  | Rule      | Hybrid    | Rule          | Hybrid        | Rule  | Hybrid |
> | o3-2025-04-16              | 23.30   | 58.90   | 28.20     | 45.05     | 53.28         | 66.95         | 25.11 | 85.02  |
> | DeepSeek-R1                | 27.55   | 55.30   | 32.05     | 40.75     | 50.43         | 61.54         | 37.44 | 79.30  |
> | claude-3-7-sonnet-thinking | 21.60   | 48.75   | 28.75     | 38.25     | 49.29         | 58.40         | 44.49 | 81.94  |
> | QwQ-32B                    | 20.65   | 53.30   | 28.55     | 39.35     | 50.43         | 63.53         | 47.58 | 75.37  |
> | DeepSeek-V3                | 22.45   | 47.05   | 27.80     | 36.05     | 46.15         | 54.42         | 44.05 | 66.52  |
> | gpt-4.1                    | 21.30   | 46.75   | 25.35     | 36.40     | 48.72         | 56.98         | 23.79 | 75.33  |
>
> Our two main contributions compared to existing test sets are a **balanced subject and difficulty distribution** and **high-quality data**.
>
> **Balanced subject and difficulty distribution.** From the results shown in the tables, we can observe that our test set ensures a **balanced distribution** not only in terms of **physics subfields** but also in **difficulty levels**, which span high school, high school competitions, undergraduate, and physics major (graduate-level) problems. Both aspects are crucial for comprehensive evaluation.
>
> As shown in Table 7 of the paper, model performance varies significantly across different subfields.  A balanced subject distribution ensures average scores better reflect overall physics ability. Similarly, a balanced difficulty distribution assesses both reasoning and knowledge. As discussed in Section 4.1, High school and Olympiad problems test reasoning, while undergraduate and graduate problems focus on content knowledge. Covering all difficulty levels enables a thorough evaluation.
>
> **High-quality data.** Among existing datasets, UGPhysics is most similar to ours but with significantly lower quality. From 2,000 UGPhysics samples, 789 questions were unanswered by six models (Table 2.3). Manual review found only **19.39%** valid, **52.75%** flawed questions, and **27.88%** flawed answers. **These issues are well addressed in our dataset.**
>
> Common question issues include: (1) Reliance on previous context (i.e., those commonly omitted in standard physics problems), (2) Missing key conditions (i.e., those commonly omitted in standard physics problems), (3) Extraction errors (e.g., truncation, missing symbols, incoherent text), (4) References to unavailable figures.
>
> Answer issues include: (1) Incomplete responses, (2) Incorrect final answers or explanations, (3) Invalid reasoning in open-ended questions.
>
> In short, our dataset’s higher quality ensures more trustworthy model evaluation results. A more detailed analysis will be presented in the next version of the paper.
>
> In summary, compared to existing test sets, ours better captures the multifaceted nature of a model’s physics proficiency. It is not merely an incremental addition but a well-constructed and comprehensive benchmark dataset.
>
> ## Weakness 2. Given this dataset as the training set, can the general reasoning ability of LRMs be improved a lot?
>
> A2. Thank you for your question. Table 8 in the paper illustrates how our data enhances models’ physical and mathematical reasoning abilities. We also evaluated Qwen-2.5 models trained on our data using GPQA-Diamond and MMLU-Pro benchmarks.
>
> Results show all 3B, 7B, and 14B parameter models improved after fine-tuning with our data, demonstrating its effectiveness in boosting general reasoning. We also observed that larger models tended to show greater improvements, likely due to their higher capacity.
>
> | Model                    | GPQA-Diamond | MMLU-pro |
> | ------------------------ | ------------ | -------- |
> | Qwen2.5-3B-Instruct      | 30.37        | 39.39    |
> | Qwen2.5-3B-Instruct-SFT  | 30.89        | 39.95    |
> | Qwen2.5-7B-Instruct      | 33.46        | 52.68    |
> | Qwen2.5-7B-Instruct-SFT  | 39.08        | 54.11    |
> | Qwen2.5-14B-Instruct     | 39.65        | 58.30    |
> | Qwen2.5-14B-Instruct-SFT | 49.62        | 65.77    |

---

> > ### Comment · Reviewer_xHz6 · 2025-08-07
> > **Thanks for the rebuttal**
> >
> > The new experimental results and discussion have solved most of my concerns. I will keep my score as it is positive. Thank you for the effort of the authors.

---

> > > ### Author Response · Authors · 2025-08-09
> > >
> > > Thank you very much for your thoughtful follow-up and for considering our response. We truly appreciate your time, the constructive feedback, and your decision to keep your score positive.

---

### Official Review · Reviewer_mXx6 · 2025-07-04

**Rating:** 5
**Confidence:** 4

**Summary:**

The authors introduce PHYSICS, a large bilingual (English/Chinese) dataset of 16,568 physics problems from five domains (Mechanics, Electromagnetism, Thermodynamics, Optics, Modern Physics) and four difficulty levels (high school to graduate). They extract questions via OCR and LLMs, verify them with experts, filter for pretraining leakage, and split into 14,568 training and 2,000 test items. To assess model answers, they propose a Rule+Model framework combining physics rules with a fine-tuned judge model (physics-xVerify). Experiments over 28 LLMs show large reasoning gaps, and fine-tuning on PHYSICS improves both physics and related math benchmark performance.

**Additional Feedback:**

Suggestion: The paper does not specify the license under which the PHYSICS dataset is released. For a dataset intended to serve as a long-term benchmark and training resource, clearly stating the usage license is essential for transparency and reusability.

**Dataset Code Accessibility:**

Partly

**Dataset Code Comments:**

Only the test split (PHYSICS_test.jsonl) is publicly available; the training split will be released later. This limits full dataset use.
No license is specified for the dataset, leaving users uncertain about reuse permissions.

**Ethical Comments:**

All problems derive from publicly available textbooks; no personal or sensitive data is used. Mathematical and physical content is factual and not subject to personal bias. When using textbook material in research, one should paraphrase original wording or cite the source to avoid any residual copyright risk.

**Ethical Considerations:**

No, there are no or only very minor ethics concerns

**Final Justification:**

New experimental results and discussion have addressed my concerns. I will keep the score as it is positive.

**Limitations Weaknesses:**

1. **Translation workflow is under-specified, and the release is text-only so far:**
The paper notes that all questions are translated, yet it does not specify the model version, decoding settings, or the fraction of human post-editing. Without these details, it is hard to judge whether translation shifts meaning or difficulty. Moreover, the dataset currently lacks diagrams and apparatus photos that often appear in physics problems, though the author illustrated in limitations. I recommend adding a “Translation” subsection describing the machine-human pipeline and its metrics, and expanding the dataset to include multimodal items in a future release.

2. **Lack of intuitive result presentation and insufficient cross-benchmark comparison:** The paper presents key performance results only in tables and line charts, which makes it difficult for readers to immediately grasp the performance gaps across models, subfields, and difficulty levels. I strongly recommend visualizing the hybrid accuracy results using grouped bar charts—first by model, and then broken down by subject area (e.g., mechanics, thermodynamics) or difficulty tier. This would help make performance disparities more salient. In addition, while the dataset is compared conceptually to other physics benchmarks (e.g., OlympiadBench, UGPhysics), the paper does not provide a visual or quantitative comparison of model performance across these datasets. Incorporating a cross-dataset bar plot that shows PHYSICS results alongside these existing benchmarks would significantly strengthen the paper’s empirical positioning and enable more meaningful contextualization.

**Strengths Contributions:**

1. **Dataset scale, tiered difficulty, and bilingual alignment jointly enable varied research:** The dataset contains 16,568 physics questions that span mechanics, electromagnetism, thermodynamics, optics, and modern physics. Difficulty labels cover four levels from senior high school to first-year graduate courses. Every item has a sentence-aligned Chinese and English version, and the authors use a 7 : 1 train/test split. This design lets researchers probe model behavior across sub-domains, difficulty levels, and languages with a single resource.

2. **Transparent pipeline with layered quality checks makes the data trustworthy:**
Starting from PDF harvesting, the authors convert to Markdown, extract question-answer pairs with GPT-4o while checking original positions, fix OCR errors, filter noise, and finally have domain experts review each sample. N-gram leakage screening is applied at multiple stages. Such a multi-stage procedure reduces the chance that question text, answers, or difficulty tags drift during processing, which is essential when later work will rely on the provided reasoning traces.

---

> ### Author Rebuttal · Authors · 2025-07-31
>
> ## Weakness 1. Details of the translation process.
>
> A1. Thank you for your question. Below is a detailed explanation of our translation process, which will be added to the data processing section of the next paper version. As noted in Section 3.2.2, after collecting 8,284 mixed Chinese-English samples, **we used GPT-4o for mutual translation**. The prompt is shown below:
>
> ```Plain
> Please act as an expert in physics and translation. Your task is to translate the Chinese/English text I provide into English/Chinese.
> The text will be a physics question or an answer, and your translation should reflect it accurately.
> If there are LaTeX expressions, numbers, or units, do not translate them.
> Adhere strictly to the meaning of the original text. Do not add or remove any content.
> Only output the translated version of the original text.
> The content you need to translate is: {{text}}
> Translation:
> ```
>
> We ensured translation quality through **LLM and human evaluations**. Gemini-2.5-Flash first assessed each translation using [1], focusing on accuracy, fluency, style, and terminology; low-quality outputs were retranslated. Then, over five experts reviewed them using the MQM framework [2], with manual corrections. This yielded 8,284 high-quality translations, doubling the dataset to 16,568 samples.
>
> ##  Weakness 2. Multimodal Version of the Data
>
> A2. Thank you for your suggestion. We are currently developing multi-modal physical data and plan to release a multi-modal version of our dataset in the future.
>
> ## Weakness 3. Lack of intuitive result presentation
>
> A3. Thank you for the suggestion. Appendix F has bar charts of model performance by language, subject, and difficulty. We plan to expand this analysis later. Since images can’t be included here, results are shown in tables.
>
> - Table 3.1 shows average accuracy (subset), revealing clear gaps between open- and closed-source models.
>
> Table 3.1 Accuracy of several models (subset).
>
> | Model                      | Rule Acc | Hybrid Acc |
> | -------------------------- | -------- | ---------- |
> | Grok3(think)               | 25.10     | 46.40       |
> | Claude 3.7 Sonnet Thinking | 21.60     | 48.75      |
> | o3-2025-04-16              | 23.30     | 58.90       |
> | GPT-4.1                    | 21.30     | 46.75      |
> | QwQ-32B                    | 20.65    | 53.30       |
> | DeepSeek-R1                | 27.55    | 55.30       |
>
> - Table 3.2 shows subject-wise performance (subset), with lower scores in Modern Physics and Thermodynamics indicating higher difficulty.
>
> Table 3.2 Model performance across subjects (subset).
>
> | Model                      |       |       | Subject     |         |        |
> | -------------------------- | ----- | ----- | ----------- | ------- | ------ |
> |                            | Modern Physics  | Mechanics | Electromagnetism | Thermodynamics | Optics |
> | Grok3(think)               | 40.75 | 54.00    | 56.75       | 35.00      | 45.50   |
> | Claude 3.7 Sonnet Thinking | 44.75 | 51.50  | 53.25       | 38.50   | 55.75  |
> | o3-2025-04-16              | 48.75 | 62.75 | 66.75       | 56.50    | 59.75  |
> | GPT-4.1                    | 43.00    | 50.50  | 55.25       | 37.75   | 47.25  |
> | QwQ-32B                    | 47.50 | 59.50  | 57.25       | 45.75   | 56.50   |
> | DeepSeek-R1                | 48.00    | 61.75 | 61.00          | 46.50    | 59.25  |
>
> - Table 3.3 shows performance by difficulty (subset), where lower scores on Olympiad and Undergraduate/Postgraduate questions reveal gaps in reasoning and domain knowledge.
>
> Table 3.3 Model performance across difficulty levels (subset). Difficulty levels include: 1: High School and Below, 2: High School Olympiad, 3: Undergraduate (Non-Physics Major), 4: Undergraduate/Postgraduate(Physics Major).
>
> | Model                      | Difficulty Level |       |       |       |
> | -------------------------- | ---------------- | ----- | ----- | ----- |
> |                            | 1     | 2     | 3     | 4     |
> | Grok3(think)               | 56.25 | 54.79 | 55.24 | 42.28 |
> | Claude 3.7 Sonnet Thinking | 68.75| 53.57 | 58.85 | 44.17 |
> | o3-2025-04-16              | 87.50 | 71.81 | 73.10  | 52.06 |
> | GPT-4.1                    | 87.50| 60.64 | 55.95 | 41.03 |
> | QwQ-32B                    | 85.42| 56.12 | 68.18 | 47.09 |
> | DeepSeek-R1                | 83.33 | 59.69 | 67.46 | 49.85 |
>
> ##  Weakness 4.insufficient cross-benchmark comparison
>
> A4. Thank you for your suggestion. We compare evaluation results on plain-text physics questions from UGPhysics, OlympiadBench, and GPQA. To reduce testing costs and better evaluate physical reasoning, UGPhysics samples 2,000 instances by topic, language, and level, preserving the original distribution (all other references to UGPhysics in the rebuttal and the main paper refer to the full dataset). Only physics questions from GPQA and OlympiadBench are used. This version includes models that performed well on PHYSICS and UGPhysics; a full comparison will appear in the next version. Results are in Table 4.1.
>
> Table 4.1 cross-benchmark comparison.
>
> | model                      | PHYSICS | PHYSICS | UGPhysics | UGPhysics | OlympiadBench | OlympiadBench | GPQA  | GPQA   |
> | -------------------------- | ------- | ------- | --------- | --------- | ------------- | ------------- | ----- | ------ |
> |                            | Rule    | Hybrid  | Rule      | Hybrid    | Rule          | Hybrid        | Rule  | Hybrid |
> | o3-2025-04-16              | 23.30   | 58.90   | 28.20     | 45.05     | 53.28         | 66.95         | 25.11 | 85.02  |
> | DeepSeek-R1                | 27.55   | 55.30   | 32.05     | 40.75     | 50.43         | 61.54         | 37.44 | 79.30  |
> | claude-3-7-sonnet-thinking | 21.60   | 48.75   | 28.75     | 38.25     | 49.29         | 58.40         | 44.49 | 81.94  |
> | QwQ-32B                    | 20.65   | 53.30   | 28.55     | 39.35     | 50.43         | 63.53         | 47.58 | 75.37  |
> | DeepSeek-V3                | 22.45   | 47.05   | 27.80     | 36.05     | 46.15         | 54.42         | 44.05 | 66.52  |
> | gpt-4.1                    | 21.30   | 46.75   | 25.35     | 36.40     | 48.72         | 56.98         | 23.79 | 75.33  |
>
> Model rankings are consistent across datasets, with GPQA scoring highest, likely due to its easier multiple-choice format. Our main contributions are a **balanced subject-difficulty distribution** and **high-quality data**.
>
> **Balanced subject and difficulty distribution.** PHYSICS and UGPhysics have similar average scores, but PHYSICS provides more balanced coverage (Tables 4.2 and 4.3). Given the performance variation across subjects and difficulty levels in Table 7 in the main paper, the average results on PHYSICS better reflect a model’s overall physics capability compared to UGPhysics.
>
> Table 4.2 The subject distribution of our PHYSICS test set. Subjects are divided Mod.(Modern Physics), Mech. (Mechanics), Electromag.(Electromagnetism), Thermo. (Thermodynamics), and Optics.
>
> | Benchmark     | Mod. | Mech. | Electromag. | Thermo. | Optics. | Total |
> | ------------- | ---- | ----- | ----------- | ------- | ------- | ----- |
> | MMLU          | 114  | 177   | 94          | 52      | 51      | 488   |
> | GPQA          | 191  | 10    | 15          | 4       | 7       | 227   |
> | Olympiadbench | 79   | 136   | 52          | 71      | 13      | 351   |
> | UGPhysics     | 4998 | 3430  | 1148        | 744     | 720     | 11040 |
> | PHYSICS(ours) | 400  | 400   | 400         | 400     | 400     | 2000  |
>
> Table 4.3 The difficulty distribution of our PHYSICS test set. Difficulty levels include: 1: High School and Below, 2: High School Olympiad, 3: Undergraduate (Non-Physics Major), 4: Undergraduate/Postgraduate(Physics Major).
>
> | Benchmark     | 1    | 2    | 3    | 4    | Total |
> | ------------- | ---- | ---- | ---- | ---- | ----- |
> | MMLU          | 386  | 0    | 102  | 0    | 488   |
> | GPQA          | 0    | 0    | 41   | 186  | 227   |
> | Olympiadbench | 0    | 351  | 0    | 0    | 351   |
> | UGPhysics     | 0    | 0    | 1463 | 9577 | 11040 |
> | PHYSICS(ours) | 48   | 196  | 418  | 1338 | 2000  |
>
> **High-quality data.** Among existing datasets, UGPhysics is most similar to ours but with significantly lower quality. From 2,000 UGPhysics samples, 789 questions were unanswered by six models (Table 4.1). Manual review found only **19.39%** valid, **52.75%** flawed questions, and **27.88%** flawed answers. **These issues are well addressed in our dataset.**
>
> Common question issues include: (1) Reliance on previous context (i.e., those commonly omitted in standard physics problems), (2) Missing key conditions (i.e., those commonly omitted in standard physics problems), (3) Extraction errors (e.g., truncation, missing symbols, incoherent text), (4) References to unavailable figures.
>
> Answer issues include: (1) Incomplete responses, (2) Incorrect final answers or explanations, (3) Invalid reasoning in open-ended questions.
>
> In short, our dataset’s higher quality ensures more reliable evaluations. A detailed analysis will appear in the next version.
>
> ##  Feedback 1. Data Copyright, Open-Source Plan, and Data License
>
> A5. Thank you for your suggestion. All data used have purchased rights or proper open licenses, and sources will be clearly annotated in the release. The test set is released, and the training set will follow after review. We plan to use the GPL license to support community, with details provided upon release.
>
> ## Reference：
>
> [1] Kocmi, Tom, and Christian Federmann. "GEMBA-MQM: Detecting Translation Quality Error Spans with GPT-4." Proceedings of the Eighth Conference on Machine Translation. 2023.
>
> [2] Lommel, Arle, Hans Uszkoreit, and Aljoscha Burchardt. "Multidimensional quality metrics (MQM): A framework for declaring and describing translation quality metrics." *Tradumàtica* 12 (2014): 0455-463.

---

> > ### Comment · Reviewer_mXx6 · 2025-08-09
> >
> > Thanks for the authors' response. Your new experimental results and discussion have addressed my concerns. I will keep the score as it is positive. Looking forward to seeing the multimodal version of the data.

---

### Note · Authors · 2025-08-13

We thank all the reviewers and Area Chairs for the time and effort they dedicated to reviewing our work. We particularly appreciate the reviewers’ recognition of our key contributions:

- **Large-scale, balanced difficulty and discipline coverage:** We proposed large-scale, multi-disciplinary, multi-difficulty training and test sets to improve and evaluate large models’ physics capabilities. (Reviewers mXx6, xHz6, ChAs, LXLB)
- **High data quality:** We developed an effective data processing pipeline, with quality inspection combining LLMs and human experts. (Reviewers mXx6, ChAs, LXLB)
- **Targeted evaluation method:** We identified the limitations of current evaluation methods in the physics domain and proposed a tailored rule+model approach. (Reviewers xHz6, LXLB)
- **Extensive experiments:** We conducted extensive experiments to validate our training and test sets’ effectiveness. (Reviewers xHz6, ChAs, LXLB)

During rebuttal, we addressed the following concerns:

- **Cross-benchmark comparison:** We detailed the advantages of our training and test sets, highlighting their large scale, broad coverage, high quality, and balanced distribution. (Reviewers mXx6, xHz6, ChAs)
- **Pipeline details:** We gave a detailed description of our translation and training processes. (Reviewers mXx6, LXLB)
- **Improvement in general reasoning ability:** We validated the improvements on general reasoning ability by improvements on GPQA-Diamond and MMLU-pro achieved by models trained on our dataset. (Reviewer xHz6)
- **Model diversity:** We used two models from the LLaMA series, two from the Mistral series, and three Qwen series models described in the main text, and their performance gains after training demonstrate the broad generalizability of our training set. (Reviewer ChAs)
- **Comparisons with training methods**: We showed that models trained on our dataset outperformed SimpleRL, General-Reasoner, and Absolute-Zero, confirming the dataset’s contribution to improved physics capabilities. (Reviewer ChAs)
- **Data deduplication:** In the main text, we performed embedding-based deduplication to remove most duplicates. In addition, we applied the method described in Openmathinstruct-2, which identified 32 duplicates out of 16,568 entries to be rewritten or removed. (Reviewer LXLB)

We are pleased all reviewers agreed our responses addressed their concerns, and we will include these results, discussions, and feedback in the next revision’s main text and appendix.

---

### Decision · Program_Chairs · 2025-09-18

**Decision:**

Accept (poster)

**Comment:**

This work introduces PHYSICS, a dataset with16,568 high-quality physics problems spanning subjects and difficulty levels. It also contributed an improved evaluation framework combining rule- and model-based judgments to advance LLMs' physical reasoning.

Overall, reviewers are positive about this work, as it contributes a large-scale dataset with balanced difficulty and diverse discipline coverage. With their data processing pipeline, the resulting corpus is of high data quality, and the authors have conducted comprehensive experiments to validate the effectiveness of the proposed dataset.  Reviewers raised some concerns about the translation workflow in the data construction process, insufficient cross-benchmark comparison, overlap with existing corpus, comparison with reasoning models, and data contamination issues.

Most of these weaknesses have been addressed in the rebuttal phase where the authors have provided detailed experiments about the cross-benchmark comparison, pipeline details, comparisons with suggested models as well as deduplication efforts.  The work has improved a lot during the discussion stage, and I recommend the authors to incorporate these additional results and discussions into their revised version.